# Modeling liquid water transport in snow under rain-on-snow conditions – considering preferential flow

Sebastian Würzer,[1,2*] Nander Wever,[2,1] Roman Juras,[1,3] Michael Lehning,[1,2] and Tobias Jonas,[1]

[1]WSL Institute for Snow and Avalanche Research SLF, Flüelastrasse 11, 7260 Davos Dorf, Switzerland

[2]École Polytechnique Fédérale de Lausanne (EPFL), School of Architecture, Civil and Environmental Engineering, Lausanne, Switzerland

[3]Faculty of Environmental Sciences, Czech University of Life Sciences Prague, Kamýcká 129, 165 21, Prague, Czech Republic

*Correspondence to*: Sebastian Würzer (sebastian.wuerzer@slf.ch)

**Abstract.** Rain-on-snow (ROS) has the potential to generate severe floods. Thus, precisely predicting the effect of an approaching ROS event on runoff formation is very important. Data analyses from past ROS events have shown that a snowpack experiencing ROS can either release runoff immediately or delay it considerably. This delay is a result of refreeze of liquid water and water transport, which in turn is dependent on snow grain properties but also on the presence of

structures such as ice layers or capillary barriers. During sprinkling experiments, preferential flow was found to be a process that critically impacted the timing of snowpack runoff. However, current one-dimensional snowpack models are not capable of addressing this phenomenon. For this study, the detailed physics-based snowpack model SNOWPACK is extended with a water transport scheme accounting for preferential flow. The implemented Richards´ Equation solver is modified using a dual-domain approach to simulate water transport under preferential flow conditions. To validate the presented approach, we

used an extensive dataset of over 100 ROS events from several locations in the European Alps, comprising meteorological and snowpack measurements as well as snow lysimeter runoff data. The model was tested under a variety of initial snowpack conditions, including cold, ripe, stratified and homogeneous snow. Results show that the model accounting for preferential flow (PF) demonstrated an improved overall performance, where in particular the onset of snowpack runoff was captured better. While the improvements were ambiguous for experiments on isothermal wet snow, they were pronounced for

experiments on cold snowpacks, where field experiments found preferential flow to be especially prevalent.

**Keywords**: *snow cover, water transport, snowpack runoff, mountain hydrology, preferential flow, rain-on-snow, one-dimensional snow model*

## 1 Introduction

The flooding potential of Rain-on-snow (ROS) events has been reported for many severe floods in the US (Kattelmann, 1997; Kroczynski, 2004; Leathers et al., 1998; Marks et al., 2001; McCabe et al., 2007), but also in Europe (Badoux et al., 2013; Freudiger et al., 2014; Rössler et al., 2014; Sui and Koehler, 2001; Wever et al., 2014b) where for example up to 70% of peak flow events could be attributed to ROS events for Austria (Merz and Blöschl, 2003). With rising air temperature due to climate change, the frequency of ROS is likely to increase in high elevation areas (Surfleet and Tullos, 2013) as well as in high latitudes (Ye et al., 2008). Besides spatial heterogeneity of the snowpack and uncertainties in meteorological forcing, deficits in process understanding make the consequences of extreme ROS events very difficult to forecast (Badoux et al., 2013; Rössler et al., 2014). For hydro-meteorological forecasters, it is particularly important to know a priori how much and when snowpack runoff is to be expected. Particularly, a correct temporal representation of snowpack processes is crucial to identify whether the presence of a snowpack will attenuate or amplify the generation of catchment-wide snowpack runoff. Most studies investigating ROS only consider the generation of snowpack runoff on a daily or multi-day timescale, where an exact description of water transport processes is less important than for sub-daily time scales (Wever et al., 2014a). Water transport processes are further usually described for snowmelt conditions, but not for ROS conditions, where high rain intensities may fall onto a cold snowpack below the freezing point. In this study however, we particularly focus on snowpack runoff generation at sub-daily scales with special attention to the timing of snowpack runoff which is influenced by preferential flow.

Many studies have shown that flow fingering or preferential flow is an important water transport mechanism in both laboratory experiments (Hirashima et al., 2014; Katsushima et al., 2013; Waldner et al., 2004) as well as under natural conditions, using dye tracer (Gerdel, 1954; Marsh and Woo, 1984; Schneebeli, 1995), temperature investigations (Conway and Benedict, 1994) or by measuring the spatial variability of snowpack runoff (Kattelmann, 1989; Marsh and Pomeroy, 1993, 1999; Marsh and Woo, 1985). The variability of snowpack runoff is defined by the distribution and size of preferential flow paths (PFP), which are dependent on the structure of the snowpack and weather conditions (Schneebeli, 1995). Beyond its importance for hydrological implications, preferential flow may also be crucial for wet snow avalanche formation processes, where snow stability can be depending on the exact location of liquid water ponding (Wever et al., 2016a).

Most snow models describe the water flow in snow as a uniform wetting front, thereby implicitly only considering the matrix flow component. The history of quantitative modeling of water transport in snow starts with Colbeck (1972), who first described a gravity drainage water transport model for isothermal, homogeneous snow. This was done by applying the general theory of Darcian flow of two-fluid phases flowing through porous media, neglecting capillary forces. Because water transport is not just occurring in isothermal conditions and snow can therefore not be treated as a classical porous medium, Illangasekare et al. (1990) were the first to introduce a 2D model being able to describe water transport in subfreezing and layered snow. A detailed multi-layer physics based snow model, where water transport was governed by the gravitational part of Richards` Equation described in Colbeck (1972) was introduced by Jordan (1991). With the implementation of the

full Richards` Equation described by Wever et al. (2014a), the influence of capillary forces on the water flow was firstly represented in an operationally used snowpack model.

A model accounting for liquid water transport through multiple flow paths was developed by Marsh and Woo (1985), but not being able to explicitly account for structures like ice layers and capillary barriers. Recently, multi-dimensional water transport models were developed, which allow for the explicit simulation of PFP (Hirashima et al., 2014). These models are valuable for describing spatial heterogeneities and persistence of PFP, but have not yet been shown to be suitable for hydrological or operational purposes. In general, multi-dimensional models are limited by the fact that they are computationally intensive, thus not thoroughly validated for seasonal snowpacks and yet lack the description of crucial processes such as snow metamorphism and snow settling.

In snowpack models which are used operationally, PFP are not yet considered. The recently introduced Richards' Equation solver for SNOWPACK led to a significant improvement of modelled sub-daily snowpack runoff rates. For this paper, we further modified the transport scheme for liquid water by implementing a dual-domain approach to represent PFP. This new approach is validated against snow lysimeter measurements which were recorded during both natural and artificial ROS events.

The study aims to better describe snowpack runoff processes during ROS events within snowpack models that can be used for operational purposes such as avalanche warning and hydrological forecasting. This requires that the model results remain reliable, i.e. that improvements are not realized on the expense of a decreased model performance during periods without ROS, and that the model must not be too computationally expensive. This is the first study to test a water transport scheme accounting for preferential flow which has been implemented in a snowpack model that meets the above requirements.

Our analysis of simulations of over 100 ROS events targets the following research questions:

- Is snowpack runoff during ROS in a 1D model better reproduced with a dual domain approach to account for preferential flow than with traditional methods considering matrix flow only?
- Are there certain snowpack or meteorological conditions, for which the performance specifically benefits if preferential flow is represented in the model?

This paper is structured as follows: Section 2 describes the snowpack model setup, the water transport models, input data and the event definition. Results of the simulations are shown in Sect. 3. This includes data of sprinkling experiments of ROS (3.1), natural ROS events (3.2) and the validation of the model on a long-term dataset from two alpine snow measurement sites (3.3). The results will be discussed in Sect. 4, followed by the general conclusions found in Sect. 5.

## 2. Methods

All results in this study are derived from simulations with the one-dimensional physics based snowpack model SNOWPACK (Bartelt and Lehning, 2002; Lehning et al., 2002a; Lehning et al., 2002b; Wever et al., 2014a) using 3 different water transport schemes, described in Sect. 2.2. The model was applied to four experimental sites that were set up for this study in the vicinity of Davos (Sect. 2.3). These sites were maintained over two winter seasons between 2014 and 2016 where data was recorded for several natural ROS events. At the same sites, we conducted a set of 6 sprinkling experiments to simulate ROS events for given rain intensities (Sect. 2.4). Furthermore, we conducted simulations for two extensive datasets from the European Alps: Weissfluhjoch (Switzerland, 46.83° N, 9.81° E, 2536 m MSL, WSL Institute for Snow and Avalanche Research SLF (2015), abbreviated as WFJ in the following) and Col de Porte (France, 45.30° N, 5.77° E, 1325 m MSL, Morin et al. (2012), abbreviated as CDP in the following). These datasets provide meteorological input data for running SNOWPACK as well as validation data, including snowpack runoff. Both datasets have already been used for simulations with SNOWPACK (Wever et al., 2014a) and provide data over more than 10 years each.

Below, the SNOWPACK model and the different water transport models are described first, followed by the description of the field sites for ROS observation in the vicinity of Davos. Then, we detail the setup of the artificial sprinkling experiments. After summarizing the WFJ and CDP dataset, we finally present the definition of ROS events that is used in this study. Most analyses were performed in R 3.3.0 (R Development Core Team, 2016) and figures were created with base graphics or ggplot2 (Wickham, 2009).

### 2.1 Snowpack model setup

The setup of the SNOWPACK model is similar to the setup used for simulations in Würzer et al. (2016). For all simulations, snow depth was constrained to observed values, which means that the model interprets an increase in observed snow depth at the stations as snowfall (Lehning et al., 1999; Wever et al., 2015). Because the study focuses on the event-scale and snowpack runoff is essentially dependent on the available snow, this approach was chosen such that we have the most accurate initial snow depth at the onset of the events to achieve the best comparability between the 3 water transport models. The temperature used to determine whether precipitation should be considered rain (measurements from rain gauges) or snow (from the snow depth sensors) was set to achieve best results for reproducing measured snow height for precipitation driven simulations for the Davos field sites (between 0°C and 1.0°C). For WFJ and CDP, this threshold temperature was set to 1.2°C, where mixed precipitation occurred proportionally between 0.7°C and 1.7°C. Turbulent surface heat fluxes are simulated using a Monin–Obukhov bulk formulation with stability correction functions of Stearns and Weidner (1993), as described in Michlmayr et al. (2008). At the Davos field sites (Sect. 2.3) incoming longwave radiative flux is simulated using the parameterization from Unsworth and Monteith (1975), coupled with a clear sky emissivity following Dilley and O'brien (1998), as described in Schmucki et al. (2014). For the roughness length $z_0$, a value of 0.002 m was used for all simulations at the Davos field sites and WFJ, whereas a value of 0.015 was used for CDP. The model was

initialized with a soil depth of 1.4, 2.2 and 2.14 m (for WFJ, CDP and Davos field sites, respectively) divided into layers of varying thickness. For soil, typical values for coarse material were chosen to avoid ponding inside the snowpack due to soil saturation. The soil heat flux at the lower boundary is set to a constant value of 0.06 W m$^{-2}$, which is an approximation of the geothermal heat flux.

5 **2.2 Water transport models**

The two previously existing methods for simulating vertical liquid water movement within SNOWPACK are either a simple so-called bucket approach (BA) (Bartelt and Lehning, 2002) or solving the Richards' Equation (RE), a recently introduced method for SNOWPACK (Wever et al., 2014a; Wever et al., 2014b).

The bucket approach represents liquid water dynamics by an empirically determined irreducible water content $\Theta_r$ which defines if water stays in the corresponding layer or will be transferred to the layer below. This irreducible water content varies for each layer according to Coléou and Lesaffre (1998). The Richards' Equation represents the movement of water in unsaturated porous media. Its implementation in SNOWPACK and a detailed description can be found in Wever et al. (2014a).

The preferential flow model presented in this study is based on the RE model, but follows a dual-domain approach, dividing the pore space of the snowpack into a part representing matrix flow and a part representing preferential flow. For both domains Richards' Equation is solved subsequently. The preferential flow model is described by (i) a function for determining the size of the matrix and preferential flow domain, (ii) the initiation of preferential flow (i.e., water movement from matrix flow to preferential flow) and (iii) an return flow condition from preferential flow to matrix flow.

The area of the preferential domain ($F$) is as a function of grain size (Eq. 1), which has been determined by results of laboratory experiments presented by Katsushima et al. (2013):

$$F = 0.0584 r_g^{-1.109} \tag{1}$$

where $r_g$ is grain radius (mm). $F$ is limited between 1% and 90% for reasons of numerical stability. The matrix domain is then accordingly defined as (1-$F$). Water is transferred from the matrix domain to the preferential domain if the water pressure head for a layer in the matrix domain is higher than the water entry pressure of the layer below, which can, according to Katsushima et al. (2013), also be expressed as a function of grain size. This condition is expected to be met if water is ponding on a microstructural transition (i.e. capillary barriers, ice lenses) inside the snowpack. Additionally, saturation was equalized between the matrix and the preferential domain, in case the saturation of the matrix domain exceeded the one in the preferential domain. To move water back into the matrix part, we apply a threshold in saturation of the preferential flow domain and water will flow back to the matrix domain once this threshold is exceeded. This threshold is used as a tuning parameter in the model.

Refreezing of liquid water in the snowpack is crucial for modeling water transport in subfreezing snow and may also be important for modeling preferential flow. The presented preferential flow model has also been used to simulate ice

layer formation under the presence of preferential flow by Wever et al. (2016b). Thereby, a sensitivity study on the role of refreeze in the preferential flow domain and the return flow condition from preferential flow to matrix flow was conducted. It was found that neglecting refreeze led to the best results for reproducing ice layer formation, but did not significantly affect the performance in reproducing measured hourly snowpack runoff. Therefore, refreeze in the preferential domain is

neglected in the presented study. The threshold in saturation for preferential flow (return flow condition) was also determined by the sensitivity study described in Wever et al. (2016b). While they determined a threshold in saturation of 0.1 to reproduce ice-layer observations at WFJ best, a value of 0.06 was determined to reproduce observed seasonal runoff best. We therefore used the value of 0.06. In contrast to Wever et al. (2016b), we did not set the hydraulic conductivity in soil to 0, because this can lead to an inaccurate representation of observed lysimeter runoff due to modelled ponding on soil, which

is not expected to happen on a snow lysimeter. Further details on the implementation of the PF model and its performance can be found in Wever et al. (2016b).

In summary, the PF model accelerates liquid water transport in the preferential domain by concentrating water mass in a smaller area, representing the area fraction of flow fingers in the snowpack. The saturation in the preferential domain is hence higher and unsaturated conductivity is larger. Further acceleration is achieved by disabling refreeze in the preferential

domain.

## 2.3 Davos field sites

Four field sites have been installed within an elevational range of 950 to 1950 m MSL in the vicinity of Davos, Switzerland, with one meteorological station and 3-4 snow lysimeters each (15 in total, 0.45m diameter). The meteorological stations provided most data necessary for running the SNOWPACK model and missing parameters were estimated as

described in Sect. 2.1. Lysimeters were installed at ground level with an approximate spacing of 10m horizontal distance. The lysimeters consisted of a funnel attached to a precipitation gauge buried in the ground, which monitored snowpack runoff with a tipping bucket. To block lateral inflow at the snow-soil interface, each lysimeter was equipped with a rim of 5 cm height around the inlet. The multiple snow lysimeter setups allowed analyzing the spatial heterogeneity of snowpack runoff. Snowpack properties (SWE, LWC, HS, TS) were manually measured directly before each natural ROS event so that

the initial conditions of the snowpack are known in detail. LWC was measured with the "Denoth meter", a device introduced by Denoth (1994). The onset of runoff was defined as the time when cumulative snowpack runoff (measured and simulated, respectively) has reached 1 mm.

## 2.4 Sprinkling Experiment Description

During winter 2014/15, a total of 6 artificial sprinkling experiments were performed on all four Davos field sites

described above to be able to investigate snowpack runoff generation for different snowpack properties. For each experiment, a sprinkling device was placed above a snow lysimeter, covered by an undisturbed natural snowpack, i.e. each

lysimeter was only used for one experiment. The device used for sprinkling was a refined version of the portable sprinkling device described in Juras et al. (2013) and Juras et al. (2016). The water used for sprinkling was mixed with the dye tracer Brilliant Blue FCF (concentration 0.4 g l$^{-1}$) to be able to observe PFPs within the snowpack. Sprinkling was performed in 4 bursts of 30 minutes each, interrupted by 30 minutes breaks. Sprinkling was conducted over a 2x2m plot centered above the lysimeters, and with an intensity of 24.7 mm h$^{-1}$, leading to a total of 49.4 mm artificial rain in each of the experiments. The intensities were determined by calibration experiments on lysimeters not covered by snow and are valid for a certain distance between the nozzle and the sprinkled surface and water pressure at the nozzle. Despite the fact that this value still represents a very intense ROS event, it is within range of natural ROS events and similar or much lower compared to previous studies (19 mm h$^{-1}$; Eiriksson et al. (2013); 48–100 mm h$^{-1}$; Singh et al. (1997)). For the sprinkling experiments, the exact timing of rain and intensities are known and the snowpack runoff measured at 1 minute intervals allowed precisely analyzing the performance of model simulations. Figure 1 shows a horizontal cut of a snowpack after the sprinkling experiment and a topview of the lysimeter after the snowpack was removed for cold and wet conditions, respectively. The blue color indicates where water transport took place and where sprinkled water was held by capillary forces or refrozen.

## 2.5 Extensive dataset for in-situ validation

Two long-term datasets from two study sites in the European Alps providing snow lysimeter data and high quality meteorological forcing data for running the energy balance model SNOWPACK were chosen to validate the different water transport models systematically. Datasets of both study sites used for the extensive in-situ validation are publicly available. The Col de Porte (CDP) site, located in the Chartreuse range in southeast France has been described in Morin et al. (2012) and the Weissfluhjoch site (WFJ) in the Swiss Alps has been described in Wever et al. (2015). WFJ (46.83° N, 9.81° E) is located at an elevation of 2536 m MSL and CDP (45.30° N, 5.77° E) is located at 1325 m MSL. CDP experiences a warmer climate than WFJ and as a consequence the snowpack produces snowpack runoff more often throughout the entire snow season and ROS events are more frequent than at WFJ. A multi-week snowpack builds up every winter season at CDP, but is, in contrast to WFJ, interrupted by complete melt in some years. The WFJ site is equipped with a 5 m$^2$ snow lysimeter, which measures the liquid water runoff from the snowpack. It has a 60 cm high rim to reduce lateral flow effects near the soil-snow interface (Wever et al., 2014a). CDP is equipped with both a 5 m$^2$ and a 1 m$^2$ lysimeter. Here we use data from the 5 m$^2$ lysimeter, but include data from the 1 m$^2$ lysimeter to discuss the uncertainty associated with measurements of the snowpack runoff. The studied period for WFJ is from October 1$^{st}$ 1999 to September 30$^{th}$ 2013 (14 hydrological years). Because of possible errors in the lysimeter data in the winter seasons of 1999/00 and 2004/05 as described in (Wever et al., 2014a), these data were excluded from the study. For CDP the studied period is from October 1$^{st}$ 1994 to July 31$^{st}$ 2011 (17 winter seasons) according to the data availability from the 5 m$^2$ lysimeter. The temporal resolution of lysimeter data is 1 hour for CDP and 10 minutes for WFJ. Simulation results for CDP and WFJ as well as lysimeter data for WFJ were aggregated to an hourly time scale.

**2.6 CDP+WFJ event definition**

As the number and characteristics of ROS events are strongly dependent on the event definition, special care needs to be taken to determine begin and end of a ROS event. Being interested in the temporal characteristics of snowpack runoff during ROS, we need to include the entire period from the onset of rain to the end of ROS induced snowpack runoff. Here we use an event definition according to Würzer et al. (2016) with slightly decreased thresholds to identify ROS events. According to this definition, a ROS event requires a minimum amount of 10 mm rainfall to fall within 24h on a snowpack with a height of at least 25 cm at the onset of rainfall. While the event is defined to begin once the first 1 mm of rain has fallen, the event ends once there is less than 3mm of cumulative snowpack runoff recorded within the following 5h. This definition resulted in a selection of 61 events at CDP and 40 events at WFJ. The model simulations were subsequently evaluated over a time window that extends the event length by 5 and 10 hours at the beginning and end, respectively (Fig. 2). These extended evaluation periods allowed to also investigate a possible temporal mismatch between modelled and observed snowpack runoff.

**3 Results**

**3.1 Experimental sprinkling experiments**

During the winter period 2014/15, 6 sprinkling experiments (Ex1-Ex6) were conducted on 4 different sites to be able to investigate snowpack runoff generation for different snowpack properties. With distinct differences in snowpack properties but controlled rain intensities, these experiments were expected to reveal the influence of snow cover properties and differences between the water transport models best. For all experiments, initial snow height (HS), snowpack temperature (TS), and LWC profiles were measured (Table 1 and Figure 3). According to these measurements, the snowpack conditions on which the sprinkling experiments were conducted can be separated into two cases: The first 3 experiments were conducted on dry and cold (i.e. below the freezing point) snow and will be called winter experiments. The snowpack of Ex4 and Ex5 was isothermal and in a wet state. At the onset of Ex 6 however, the snowpack was not completely isothermal and had just little LWC. Nevertheless the snowpack already passed peak SWE and was in its ablation phase. Therefore the later 3 experiments (Ex4-Ex6) will be referred to as spring experiments in the following.

For all winter experiments (Fig. 4 and Fig. 5, (a,b,c)), both modeled and observed total event runoff remained below the amount of sprinkling water. Energy input estimated by the SNOWPACK simulations suggests that snowmelt was insignificant for the winter experiments, but refreeze led to significant retention of liquid water. Additionally some sprinkled rain was retained as LWC at the end of the experiments. During Ex3 no snowpack runoff was observed, visual inspection afterwards revealed an impermeable ice layer covering both the lysimeter and the adjacent ground. During spring conditions, on the other hand, snowmelt (5.1, 8.4 and 27.4 mm respectively) led to snowpack runoff exceeding total sprinkling input, except for measured snowpack runoff in Ex6 (Fig. 4 and Fig. 5, (d,e,f)).

Additionally, Fig. 5 shows, that only the PF model was capable to reproduce all 4 peaks of observed snowpack runoff for winter conditions (Ex1+2), and even the magnitude of the first peak of Ex1 was captured well. For spring conditions however, all 3 models managed to represent 4 peaks corresponding to the four sprinkling bursts, but the PF model showed best correspondence with observed snowpack runoff (Fig. 4 and Fig. 5 (d,e,f); Table 1). Regarding the onset of snowpack runoff, the PF model especially led to faster snowpack runoff for the first 2 winter experiments, where the RE and BA models showed delayed snowpack runoff onset. For spring conditions the faster snowpack runoff response of the PF model led to a slightly early snowpack runoff. Maximal snowpack runoff rates for dry and cold conditions were generally overestimated by all models, whereas wetter conditions led to a minor underestimation (except for Ex3, where no snowpack was measured).

Regarding the overall correlation between measured and simulated snowpack runoff, PF outperformed the other models (Table 1), in particular during winter conditions. Summarizing, this initial assessment suggests that the PF approach has potential advantages in particular a) as to the timing of snowpack runoff and b) for cold snowpacks which are not yet entirely ripened.

## 3.2 Natural occurring ROS Events

In January 2015, two ROS events occurred in the vicinity of Davos. They were observed over an elevational range of 950 to 1560 m MSL on the same sites on which also the sprinkling experiments were conducted. Figure 6 shows the course of cumulative rainfall and snowpack runoff for both dates and all sites. Pre-event conditions (HS, LWC, TS) were measured shortly before the onset of rain for both events and are shown together with coefficients of determination ($R^2$) for hourly snowpack runoff of the different models Table 2.

For the event of 03.01.2015 (Fig. 6, upper row) the lower sites Serneus and Klosters (950 and 1200 m asl) showed a similar snowpack runoff dynamics regarding the delayed onset and the total amount (cumulative sum averaged over the 3 corresponding lysimeters: 20.3 mm and 21.1 mm, respectively). Also the heterogeneity between data from the individual lysimeters was relatively low (Range of 3.1 mm and 3.9 mm, respectively). For the highest located site (Davos), however, the snowpack runoff measured by all 4 lysimeters showed a greater variability (Fig. 6c) in the delayed onset of snowpack runoff (0 to 7 hours) and the total amount of snowpack runoff (mean 24.7 mm; range of 57.9 mm). The snow cover mostly built up within one week before the event. Cold temperatures led to a light melt refreeze crust at the top, but no distinct ice layers were observed. For the lower sites (Serneus and Klosters), the PF and RE models generated snowpack runoff too early (PF: approx. 3 hours; RE: 0.2 to 1.4 hours). The BA model generated snowpack runoff rather too late (1.3 to 2 hours), but still within range of the variability of observed snowpack runoff for Serneus. However, the cumulative lysimeter snowpack runoff showed good accordance with modelled PF and RE snowpack runoff at Serneus, whereas PF led to an overestimation at Klosters and BA to an underestimation of cumulative snowpack runoff at all sites. At the higher elevation site Davos, the RE model led to a better representation of mean observed snowpack runoff amount, when compared with BA and PF. The

mean observed snowpack runoff onset however was represented best by the PF model (0.3 hours early) if being compared to BA (3.4 hours delay) and RE (1.1 hours delay).

For the event of 09.01.2015 (Fig. 6, bottom row) the lower sites showed again little temporal and spatial heterogeneity in lysimeter runoff (Range of 1 mm and 2.2 mm, respectively), whereas this was more the case for Davos again (Range of 13.3 mm) probably owing to ice layers that were formed after the event on Jan 3[rd]. Observed mean event snowpack runoff was more diverse for all elevations, where Klosters had the highest cumulative snowpack runoff (Serneus 13.3 mm; Klosters 17.7 mm; Davos 7.8 mm). If compared to observed total snowpack runoff, the PF model overestimated snowpack runoff for Serneus and Klosters, whereas the RE and especially the BA model underestimate event snowpack runoff for both sites. For Davos, all models were overestimating event snowpack runoff and led to early snowpack runoff. Except the RE model, which represented onset of snowpack runoff correctly for Serneus, none of the models were able to model snowpack runoff onset correctly for any of the sites.

### 3.3 Validation on an long-term dataset

### 3.3.1 Modeled and observed snowpack runoff for the whole dataset

Given the partly contradictory findings on the performance of the three model variants based on the above assessment for artificial ROS simulations under controlled conditions (Sect. 3.1), as well as natural ROS events (Sect. 3.2), further more systematic model tests were needed. Therefore we validate the different models based on extensive datasets from the two sites WFJ and CDP, as described in Sect. 2.4.

Before we focus on the specific performance of the PF model for a large number of individual ROS events, we first analyzed the overall model performance throughout the whole study period, i.e. over entire winter seasons. For this, we analyzed observed and modeled hourly snowpack runoff provided snow heights exceeded 10 cm to ensure that lysimeter runoff was caused by snowpack runoff and not rainfall. For both sites, $R^2$ values for PF were slightly higher than for RE (Table 3), which both clearly outperformed the BA. Also the root mean squared errors (RMSE) of the PF model were lower compared to RE and BA. We can therefore conclude that the implementation of the PF approach slightly improves water transport over entire winter seasons.

### 3.3.2 ROS event characteristics of the extensive dataset

Median characteristics of the individual ROS events at CDP and WFJ are summarized in Fig. 7. The temporal course of median rain and snowpack runoff rates of all events at WFJ (40 individual events) and CDP (61 individual events) are shown in Fig. 7 (a, b). ROS events at WFJ showed generally higher maximum rain intensities than at CDP, leading to

higher median snowpack runoff intensities in the beginning of the events. Whereas at WFJ, ROS events tended to be short and intense, at CDP the event rainfall extended over a longer period of time. Interestingly, we observed relatively high initial snowpack runoff rates before the actual begin of the ROS event, especially for WFJ, which suggests that many ROS events at this site occurred during the snowmelt period. Median snowpack runoff reached a peak after 1 and 3 hours after the onset
of rain for WFJ and CDP, respectively. At WFJ snowpack runoff and rain rates in the beginning of the events were generally higher than at CDP. The course of the median air temperature during ROS events at both sites is shown in Fig. 7 (c). Especially for WFJ, median air temperature (TA) dropped with the onset of rain and median TA was higher than at CDP. The mean initial ROS event snow height (HS) for WFJ was 95 cm, which is approximately the average snow height during mid-June (for 70 years of measurements). The mean initial HS for CDP is 67 cm. With a SD of 42 cm, the variability of
initial HS for WFJ was higher than for CDP (29 cm).

### 3.3.3 Modelled and observed snowpack runoff at the event scale

Below we investigate the performance of the three water transport schemes at the event scale. Modeled snowpack runoff was assessed against observations by the coefficient of determination ($R^2$) and the root mean squared errors (RMSE). To further analyze the representation of snowpack runoff timing, we defined an absolute time lag error (TLE) as the
difference between the onsets of modelled and observed snowpack runoff in hours. The onset of snowpack runoff is defined as the time when cumulative snowpack runoff has reached 10% of total event-snowpack runoff.

Figure 8 shows boxplots of $R^2$ (a,d), RMSE (b,e) and absolute TLE (c,f) for all 40 ROS events at WFJ (a,b,c) and 61 events at CDP (d,e,f), respectively. For both sites, $R^2$ values show that the BA model performance was inferior to the RE model which was in turn slightly outperformed by the PF model. The interquartile range of $R^2$ values for CDP was generally
higher than for WFJ and increased from BA to RE, whereas it was decreasing for PF. The PF also led to a reduction in RMSE by approximately 50% if compared to the BA, but less (9% for WFJ and 25% for CDP) if compared to the RE model. Whereas the median of TLEs for all models at WFJ was 0 and therefore all models reproduced the onset of snowpack runoff very well, the interquartile range decreased from BA to the RE and PF models. The same behavior in interquartile range decrease could be observed for CDP, where the magnitude of TLE was higher than for WFJ and mostly negative. The
median TLE was again 0 for the PF and -1 h in the case of BA and RE, indicating that for these models, snowpack runoff was on average a bit delayed compared to the observations. For WFJ, TLE for BA was more often positive (early modelled snowpack runoff), which led to a very good median for BA, but also a larger interquartile range. Hence, the PF model showed the most consistent results, especially if regarding the interquartile range. For CDP we added the comparison between the 1 and 5 $m^2$ lysimeters installed at CDP (Sect. 2.5) as a reference to Fig. 8, referred to as RL. This comparison
can be seen as a benchmark performance, as it represents the measurement uncertainty of the validation dataset. As expected, RL shows the highest overall performance measures, but while the results for both PF and RE were reasonably close to those of RL, the BA model performed considerably worse.

The results shown in Fig. 8 may be influenced by both a time lag as well as the degree of reproduction of temporal dynamics. To separate both effects, we conducted a cross-correlation analysis, allowing a shift of up to 3 hours to find the best $R^2$ value. Figure 9 shows both the time lag, as well as the best $R^2$ value achieved. Interestingly, the BA model showed best correlations if the modeled snowpack runoff was shifted by 1 or 2 hours (consistently too early compared to observations). The RE model, on the other hand, showed best correlations for a shift in the other direction (consistently too late compared to observations). Neither was the case for PF with lags centered around 0.

The $R^2$ of the cross correlation analysis gives some indication of how well the temporal dynamics of the observed snowpack runoff can be reproduced, neglecting a possible time lag. The results in Fig. 9 show an improvement in $R^2$ values for both sites and all models if a time lag is applied. Greatest improvements were observed for the BA model for both sites. The good timing with the PF model is confirmed by almost no lag for WFJ and only a small lag for CDP needed to maximize $R^2$. For CDP, both RE and PF had maximized $R^2$ values in range of the lysimeter comparison (RL).

## 4 Discussion

Even though preferential flow of liquid water through snow is a phenomenon that is known and investigated since a long time, it has not yet been accounted for in 1D snow models that are in use for operational applications. The results of this study show that including this process into the water transport scheme can improve the prediction of snowpack runoff dynamics for individual ROS events as well as for the snowpack runoff of entire snow seasons. Moreover, the representation of the onset of snowpack runoff is improved. This is particularly important at the catchment scale, where a delay of snowpack runoff relative to the start of rain may affect the catchment runoff generation, especially if the time lag varies across a given catchment.

During the sprinkling experiments, sprinkling intensities were higher than average rain intensities during ROS but still within range of peak rain intensities during naturally occurring ROS events in the Swiss Alps (Rössler et al., 2014; Würzer et al., 2016) and the Sierra Nevada, California (Osterhuber, 1999). The use of the PF model clearly led to a better representation of the runoff dynamics for all experiments, including shallow and ripe snowpacks during spring conditions as well as cold and dry snowpacks representing winter conditions. The improvements were strongest for winter conditions, suggesting that under these conditions accounting for preferential flow is most relevant. This is supported by observations of preferential flow paths during winter conditions (Fig. 1 (a)), which were not visible after the spring experiments. During winter conditions just a fraction of the lysimeter area was colored with tracer, indicating preferential flow of the sprinkled water (Fig. 1 (b)), whereas spring conditions left the whole cross section of the lysimeter colored (Fig. 1 (c)). While a fast runoff response can be expected for wet and shallow snowpack and may be easier to handle for all models tested, it is the cold snowpacks that both RE and BA models did not manage to represent well: runoff from these models was more than one

hour delayed (Ex1 and Ex2), and missed approx. 10 mm of snowpack runoff within the first hour of observed runoff. This can partly be explained by the fact that BA and RE need to heat up the subfreezing snowpack before they can generate snowpack runoff, whereas refreezing is neglected in the preferential domain of the PF model and runoff can occur even in a not yet isothermal snowpack. Adjusting parameters like the irreducible water content $\Theta_r$ for the BA model could probably
lead to earlier runoff under these conditions, but thereby lead to earlier runoff, for example for WFJ events, where TLE already is positive for several events.

Despite the improved representation of the temporal runoff dynamics of the PF model (Table 1), the total event runoff of both RE and PF models is very similar for most conditions. Notably, the total event runoff for dry snowpacks is mostly overestimated by all models, suggesting an underestimation of water held in the capillarities. In cold snowpacks,
dendricity of snow grains may still be high, such that water retention curves developed for rounded grains underestimate the suction. Additionaly, high lateral flow was observed during the experiment for those conditions (Fig. 1a). This leads to an effective loss of sprinkling water per surface area of the lysimeter, which of course cannot be reproduced by the models. Therefore, observed snowpack runoff likely underestimates the snowpack runoff that would have resulted from an equivalent natural ROS event and we assume that the performance of the PF and RE models to capture the event runoff is probably
better than reported in Table 1. Note that neglecting refreeze in the PF model should not be accountable for differences in the total event runoff between the RE and PF model, if we assume that the cold content is depleted by the end of the event.

Interestingly, despite having the coldest snowpack, time lag for the 1$^{st}$ natural ROS event at Davos was shorter than for the other 2 sites. This relationship where a cold and non-ripe snowpack led to smaller lag times was also found during sprinkling experiments conducted by Juras and Würzer (unpublished data). We assume that this is an indication for the
presence of pronounced preferential flow paths under those conditions, which is also supported by the high spatial variability of snowpack runoff. Glass et al. (1989) state that the fraction of preferential flow per area is decreasing with increasing permeability, which itself was found to be increasing with porosity (Calonne et al., 2012). Therefore, with a decreasing preferential flow area due to lower densities, the cold content of a snowpack loses importance, but saturated hydraulic conductivity is reached faster within the preferential flow paths. The combination of those effects then is suspected to lead to
earlier runoff. This behavior should be ideally reproduced by the PF model and indeed the onset of runoff is caught well for this event. Here, our multi-lysimeter setup raises the awareness that the observed processes can show considerably spatial heterogeneity as e.g. documented in Figure 6. Also the formation of ice layers underlies spatial heterogeneity. Moreover the creation of preferential flow paths is strongly dependent on structural features like grain size transitions leading to capillary barriers. Unfortunately, no detailed information about grain size is available in the observations to verify this.

The PF model led to improvements in reproducing hourly runoff rates at CDP and WFJ for a dataset comprising several years of runoff measurements. This is an important finding, demonstrating that the new water transport scheme aimed at a better representation of preferential flow during ROS events, did not negatively impact on the overall robustness of the model. To the contrary, the overall performance over entire seasons could even be improved. All 3 models represent

the overall seasonal runoff better for WFJ than for CDP (Table 3), which was also found on the event scale (Fig. 8). Moreover, the CDP simulations exhibit a larger interquartile range in $R^2$ values and are therefore generally less reliable. The observed differences in model performance between both sites may either be caused by differences in snowpack or meteorological conditions or by issues with the observational data. Moreover, SNOWPACK developments have in the past
often been tested with WFJ data, which could lead to an unintended calibration favoring model applications at this site. Despite an obvious contrast in the elevation of both sites, the average conditions during ROS events seem to vary. Figure 7 suggests that at WFJ short and rather intense rain events dominate. The higher maximum rain intensities at WFJ, compared to CDP, are probably due to the later occurrence of ROS at this site (May-June), where air temperatures and therefore rain intensities are usually higher than earlier in the season (Molnar et al., 2015). Regarding mean intensities over the event scale,
data shown in Fig. 7 further imply that short and intense ROS events typically attenuate the rain input (ratio runoff to rain < 1), whereas long ROS event rather lead to additional runoff from snowmelt, which is in line with results presented in Würzer et al. (2016).

Snow height is generally higher at WFJ where the average initial snow height for the ROS events analyzed was 30 cm higher than at CDP. Ideally, the performance of the water transport scheme in the snowpack should not be affected by the
snow depth. At both sites, the snowpack undergoing a ROS event is mostly isothermal with a mean initial LWC of 1.8 vol% (CDP) and 3.0 vol% (WFJ). The initial snowpack densities at both sites were quite different. At WFJ, densities for all ROS events are around 450-500 kg m$^{-3}$, whereas for CDP densities are spread from around 200 kg m$^{-3}$ up to 500 kg m$^{-3}$. This suggests that the variable performance of all models at CDP (Fig. 8d) may be associated with early season ROS events. At CDP, a linear regression fit suggests a positive, albeit weak correlation between snowpack bulk densities and event-$R^2$ for
the RE ($R^2$ of 0.2), but no correlation for both the PF and the BA model. It seems that the RE model had some difficulties with low density snow, which was not the case for the PF model (Figure 10). This may explain why PF outperformed RE at CDP, but not for WFJ.

Remaining inaccuracies in the representation of runoff for low densities for both models applying the Richards`Equation may be explained by the fact that the water retention curve have been derived by laboratory
measurements with high density snow samples (Yamaguchi et al., 2012). Also, the parameters defining the preferential flow area (*F*) have been developed from snow samples with a density mostly above 380 kg m$^{-3}$ (Katsushima et al., 2013).

We further analysed snowpack stratigraphy derived from the SNOWPACK simulations, such as marked grain size changes (bigger 0.5 mm) and density changes (bigger 100 kg m$^{-3}$) in two adjacent simulated layers as well as the wet layer ratio (percentage of layers exceeding 1 vol% over layers below 1 vol%) and the percentage of melt forms (Table 4). These
stratigraphy measures represent possible capillary barriers having implications on the singe event-$R^2$ and might help understanding the advantages and disadvantages of the different models. Any considerable correlation between the abundance of stratigraphy features and event-$R^2$ would be indicative of potential errors in the respective model. Negative albeit small correlations could be found between the number of grain size changes and the event-$R^2$ for WFJ. Similar

correlations were noted with regards to the number of changes in density between layers for the RE and PF model. In both cases correlations were less negative for the PF model indicating a more balanced and ultimately less degraded performance with increasing number of potential capillary barriers. While at WFJ most events occurred with ripe snow this was not the case for CDP. There, positive correlations were found between the ratio of melt forms and the wet layer ratio with event-R2 for the RE model (0.33 and 0.44) and for the PF model (0.14 and 0.16). Also in this case the PF model showed more balanced results that were less influenced by the initial LWC, which is in line with our findings of the sprinkling experiments.

System input rates (sum of melt rates and rain rates) are known to significantly affect water transport processes. E.g. the area of preferential flow (Eq. 1) is likely to depend on the water supply rate. Data using sandy soils from Glass et al. (1989), shown in DiCarlo (2013), suggest that with increasing system input rates the finger width of preferential flow is increasing. Even though we have used the lowest influx rates from Katsushima et al. (2013), these rates still exceeded what seems representative of natural ROS events. We therefore analysed the effect of system input rates on the performance of our water transport models. Positive, albeit weak correlations ($R^2$ of 0.07 to 0.21) could be observed between event-$R^2$ and system input rates for all models, suggesting that they generally performed (slightly) better for higher influx rates. For the PF model this could probably be explained by the preferential flow parameters depending on laboratory measurements with high influx rates.

In combination with the hydraulic properties for lower density snow samples, additional laboratory experiments might be able to determine the number and size of preferential flow paths for lower input intensities and snow densities. Especially the calibrated parameters threshold for saturation ($\Theta_{th}$) and the number of preferential flow paths for refreeze (N) could benefit from such experimental studies. Even though CDP and WFJ provide long-term measurements on an adequate temporal resolution, this data gives little information about spatial variability of snowpack runoff limiting further validation opportunities. Large area multi-compartment lysimeter setups might help improving estimating size, amount and spatial heterogeneity of flow fingers. Sprinkling experiments with preferably low sprinkling intensities on such a device could fill a knowledge gap about water transport in snow under naturally occurring conditions.

## 5 Conclusions

A new water transport model is presented that accounts for preferential flow of liquid water within a snowpack. The model deploys a dual-domain approach based on solving the Richards' Equation for each domain separately (matrix and preferential flow). It has been implemented as part of the physics based snowpack model SNOWPACK which enables for the first time to account for preferential flow paths within a model framework that is used operationally for avalanche warning purposes and snow melt forecasting.

The new model was tested for sprinkling experiments over a natural snowpack, dedicated measurements during natural ROS events, and an extensive evaluation over 101 historic ROS events recorded at 2 different alpine long-term research sites. This assessment led to the following main conclusions:

Compared to alternative approaches, the model accounting for preferential flow (PF) demonstrated an improved
overall performance, particularly for lower densities and initially dry snow conditions. This led to smallest interquartile ranges for $R^2$ values and considerably decreased RMSE for a set of more than 100 ROS events. When evaluated over entire winter seasons, the performance statistics were superior to those of a single domain approach (RE), even if the differences were small. Both PF and RE models, however, outperformed the model using a bucket approach (BA) by a large margin (increasing median $R^2$ by 0.49 and 0.48 for WFJ and 0.53 and 0.48 for CDP). In sprinkling experiments with 30-min bursts
of rain at high intensity, the PF model showed a substantially improved temporal correspondence to the observed snowpack runoff, in direct comparison to the RE and BA models. While the improvements were small for experiments on isothermal wet snow, they were pronounced for experiments on cold snowpacks.

Model assessments for over 100 ROS events recorded at two long-term research sites in the European Alps revealed rather variable performance measures on an event-by-event basis between the three models tested. The BA model tended to
predict too early onset of snowpack runoff for wet snowpacks and a delayed onset of runoff for cold snowpacks, whereas RE was generally too late, especially for CDP. Combined with results from a separate cross correlation analysis, results suggested the PF model to provide the best performance concerning the timing of the predicted runoff.

While there is certainly room for improvements of our approach to account for preferential flow of liquid water through a snowpack, this study provides a first implementation within a model framework that is used for operational
applications. Adding complexity to the water transport module did not negatively impact on the overall performance and could be done without compromising the robustness of the model results.

Improving the capabilities of a snowmelt model to accurately predict the onset of snowpack runoff during a ROS event is particularly relevant in the context of flood forecasting. In mountainous watersheds with variable snowpack conditions, it may be decisive if snowpack runoff occurs synchronously across the entire catchment, or if the delay between
onset of rain and snowpack runoff is spatially variable e.g. with elevation. In this regard, accounting for preferential flow is a necessary step to improve snowmelt models, as shown in this study.

**Acknowledgements**

We thank the Swiss Federal Office for the Environment FOEN and the scientific exchange program Sciex-NMSch (project code 14.105) for the funding of the project. Special thanks belong to Jiri Pavlasek for making it possible to conduct the
sprinkling experiments, the extensive work and valuable exchange of ideas during the experiments. We also would like to thank Timea Mareková and Pascal Egli for their help during the experiments.

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

**Table 1: Snowpack pre-conditions and execution dates for the sprinkling experiments as well as $R^2$ values for the different model simulations. Measured values are snow height (HS), bulk liquid water content (LWC), bulk snow temperature (TS).**

| Experiment | Initial snowpack conditions | | | | $R^2$ of hourly runoff of the simulations | | |
|---|---|---|---|---|---|---|---|
| | HS [cm] | LWC [vol%] | TS [°C] | DATE | RE | PF | BA |
| Serneus (Ex1) | 48.5 | 0.1 | -1.3 | 26-Feb-15 | 0.14 | 0.59 | 0.09 |
| Davos (Ex2) | 54.5 | 0.4 | -2.5 | 27-Feb-15 | 0.24 | 0.62 | 0.08 |
| Sertig (Ex3) | 71.5 | 0 | -1.6 | 28-Feb-15 | NA | NA | NA |
| Klosters (Ex4) | 15.7 | 6.9 | 0 | 26-Mar-15 | 0.75 | 0.96 | 0.86 |
| Klosters (Ex5) | 7 | 4.9 | 0 | 8-Apr-15 | 0.70 | 0.84 | 0.88 |
| Davos (Ex6) | 39.3 | 0.9 | -0.6 | 10-Apr-15 | 0.58 | 0.83 | 0.36 |

**Table 2: Snowpack pre-conditions and $R^2$ for hourly snowpack runoff for natural events Jan 03 + Jan 09**

| | Site | Pre-event snowpack conditions | | | $R^2$ for hourly snowpack runoff | | |
|---|---|---|---|---|---|---|---|
| | | HS (cm) | LWC (vol%) | TS (°C) | RE | PF | BA |
| **03.01.2015** | **Serneus** | 19 | 0 | 0 | 0.63 | 0.35 | 0.83 |
| | **Klosters** | 24 | 0 | -0.1 | 0.72 | 0.39 | 0.78 |
| | **Davos** | 20 | 0 | -0.4 | 0.27 | 0.33 | 0.17 |
| **09.01.205** | **Serneus** | 14.5 | 0.1 | -0.2 | 0.94 | 0.57 | 0.79 |
| | **Klosters** | 18 | 0.1 | -0.2 | 0.84 | 0.73 | 0.73 |
| | **Davos** | 19.5 | 0.1 | -0.6 | 0.00 | 0.04 | 0.00 |

**Table 3: $R^2$ and mean absolute errors for hourly snowpack runoff for 17 and 14 years, for CDP and WFJ, respectively.**

|  | $R^2$ hourly snowpack runoff | | | RMSE of snowpack runoff (mm h$^{-1}$) | | |
|---|---|---|---|---|---|---|
|  | BA | RE | PF | BA | RE | PF |
| **CDP** | 0.33 | 0.50 | 0.52 | 0.56 | 0.44 | 0.40 |
| **WFJ** | 0.48 | 0.77 | 0.78 | 0.51 | 0.30 | 0.28 |

**Table 4: Correlations between event-$R^2$ and stratigraphic features at WFJ and CDP. Stratigraphic features are marked grain size changes (bigger 0.5 mm) and density changes (bigger 100 kg m$^{-3}$) in two adjacent simulated layers as well as the wet layer ratio (percentage of layers exceeding 1 vol% over layers below 1 vol%) and the percentage of melt forms. Bold numbers denote negative correlations, italic values denote positive correlations.**

| | | $R^2$ between event-$R^2$ and | | | |
| | | No. of grain size changes | No. of density changes | ratio of melt forms | wetting ratio |
|---|---|---|---|---|---|
| **WFJ** | PF | **0.19** | **0.20** | **0.03** | **0.04** |
| | RE | **0.29** | **0.22** | *0.03* | *0.02* |
| | BA | **0.31** | *0.03* | **0.01** | **0.01** |
| **CDP** | PF | **0.02** | *0.00* | *0.14* | *0.16* |
| | RE | **0.04** | *0.01* | *0.33* | *0.44* |
| | BA | **0.01** | **0.07** | *0.02* | *0.02* |

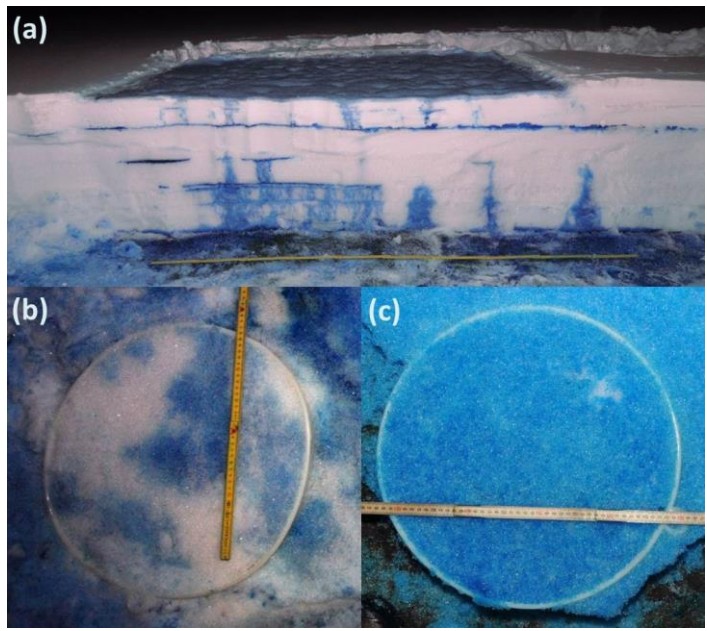

**Figure 1: (a) Horizontal cut of a snowpack after the sprinkling experiment Sertig Ex3 (26.02.2015). Lateral flow and the presence of PFP were observed. PFP were generated at regions with rain water ponding at ice layers and layer boundaries with a change in grain size (creating capillary barriers). (b) Lysimeter area after sprinkling during winter conditions (Serneus Ex1, 26.02.2015): Colored areas indicate the area where water percolated due to preferential flow. (c) Lysimeter area after sprinkling during spring conditions (Klosters Ex4, 26.03.2015): Colored area shows that water percolated uniformly, indicating dominating matrix flow.**

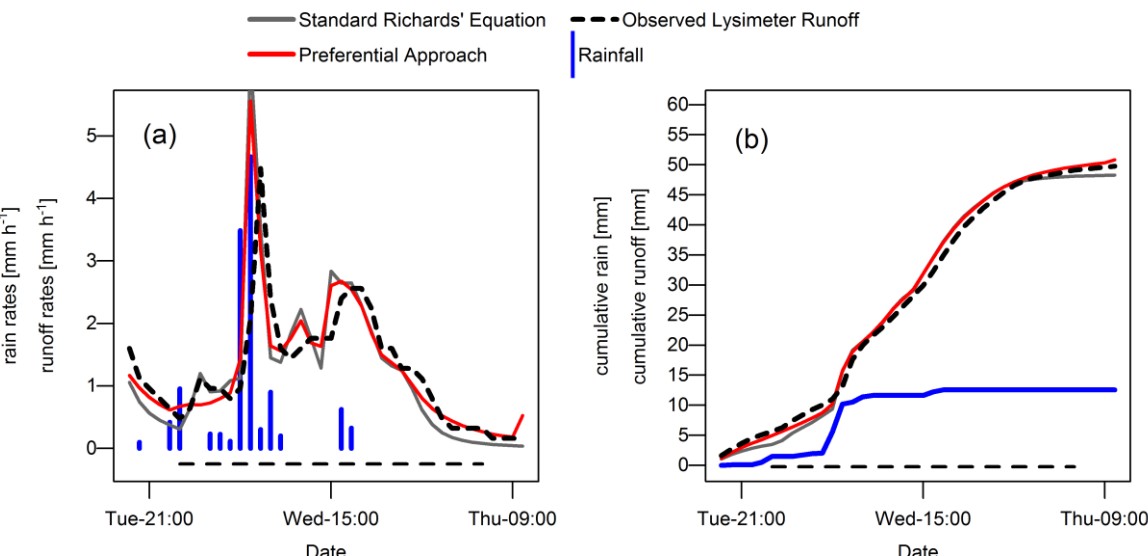

**Figure 2: Example of a ROS event occurring at WFJ. The entire extent of the x-axis refers to the evaluation period; the bar above the x-axis refers to the event length. (b) Cumulative version of the plot.**

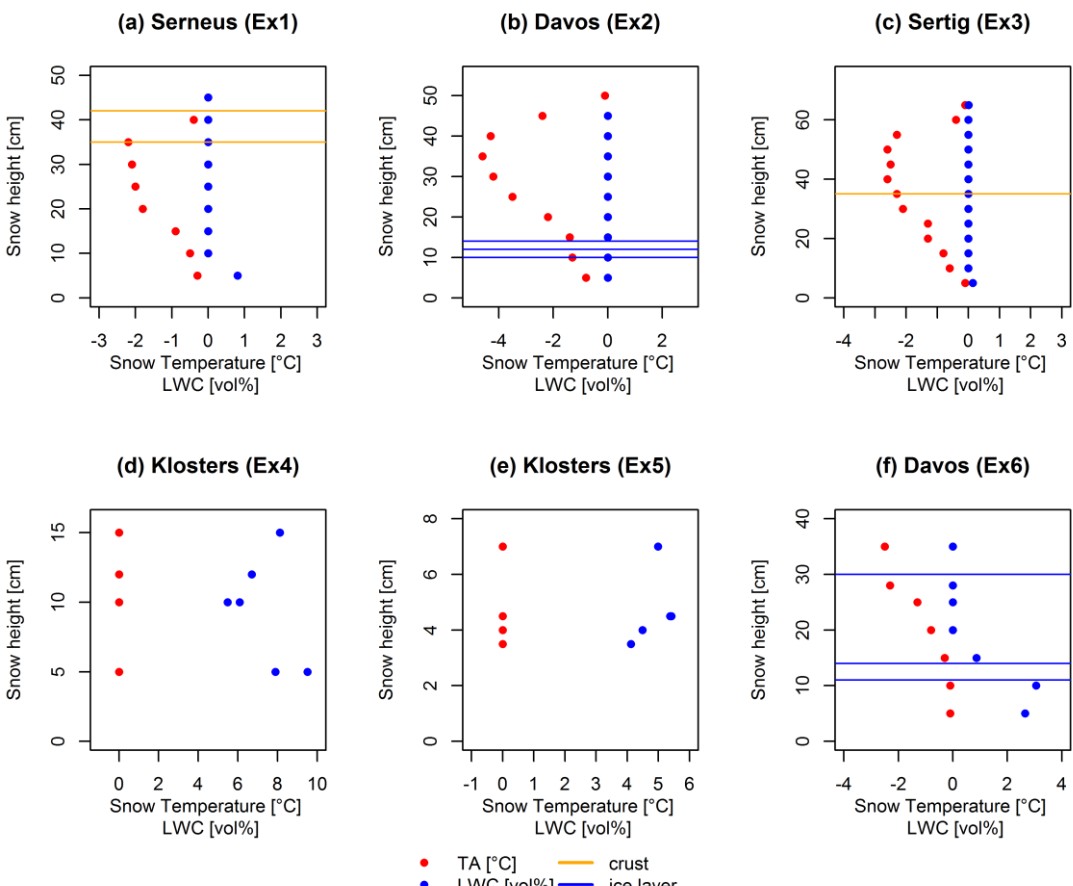

**Figure 3: Snow temperature and LWC profiles measured directly before the sprinkling experiment started. The lines represent observed ice layers (blue) and crusts (orange).**

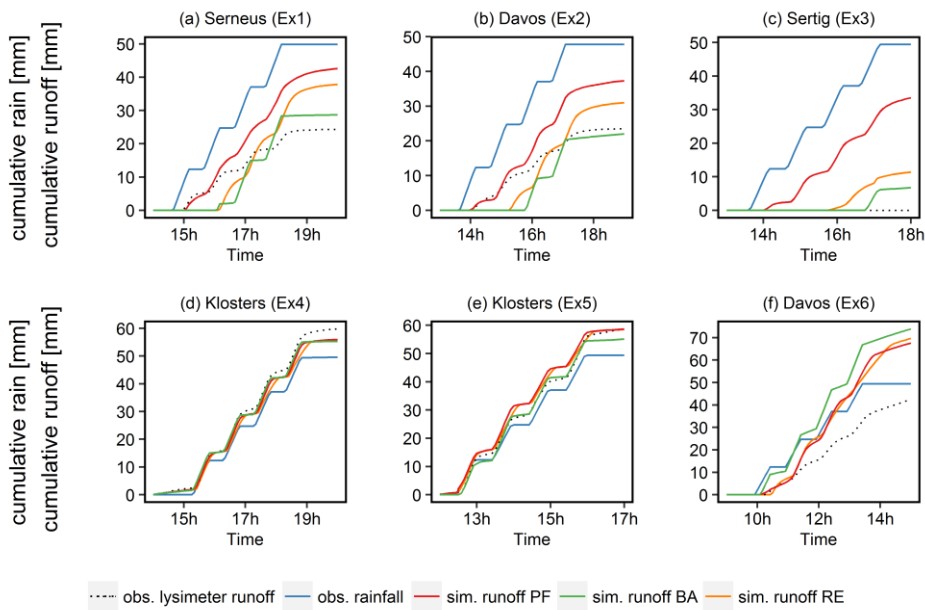

**Figure 4: Cumulative rain and snowpack runoff displayed for the six sprinkling events. Ex1 (a) - Ex3 (c) were conducted during winter conditions, Ex4 (d) – Ex6 (f) were conducted during spring conditions.**

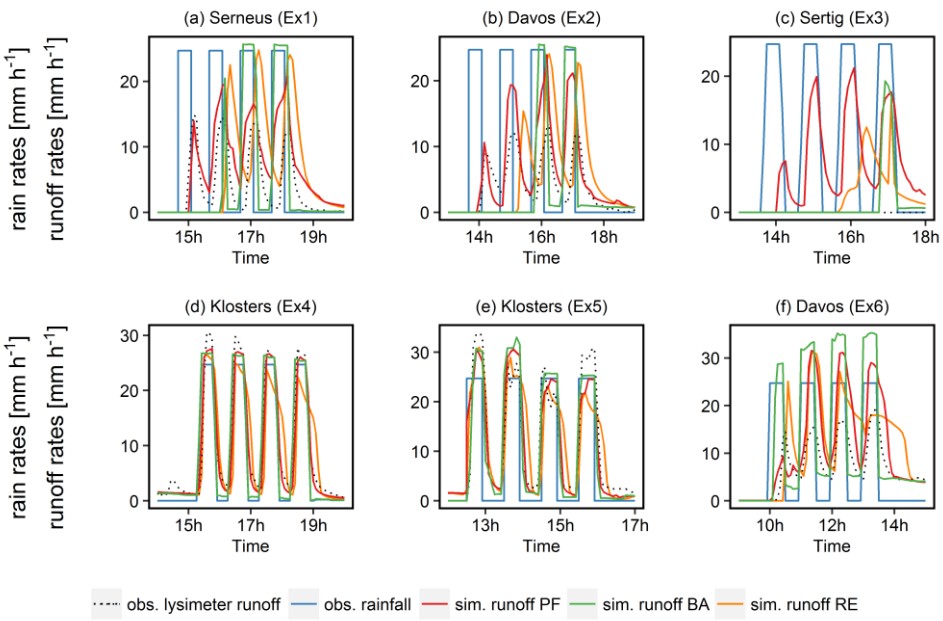

**Figure 5: Rain and snowpack runoff displayed as hydrographs for the six sprinkling events. Ex1 (a) - Ex3 (c) were conducted during winter conditions, Ex4 (d) – Ex6 (f) were conducted during spring conditions.**

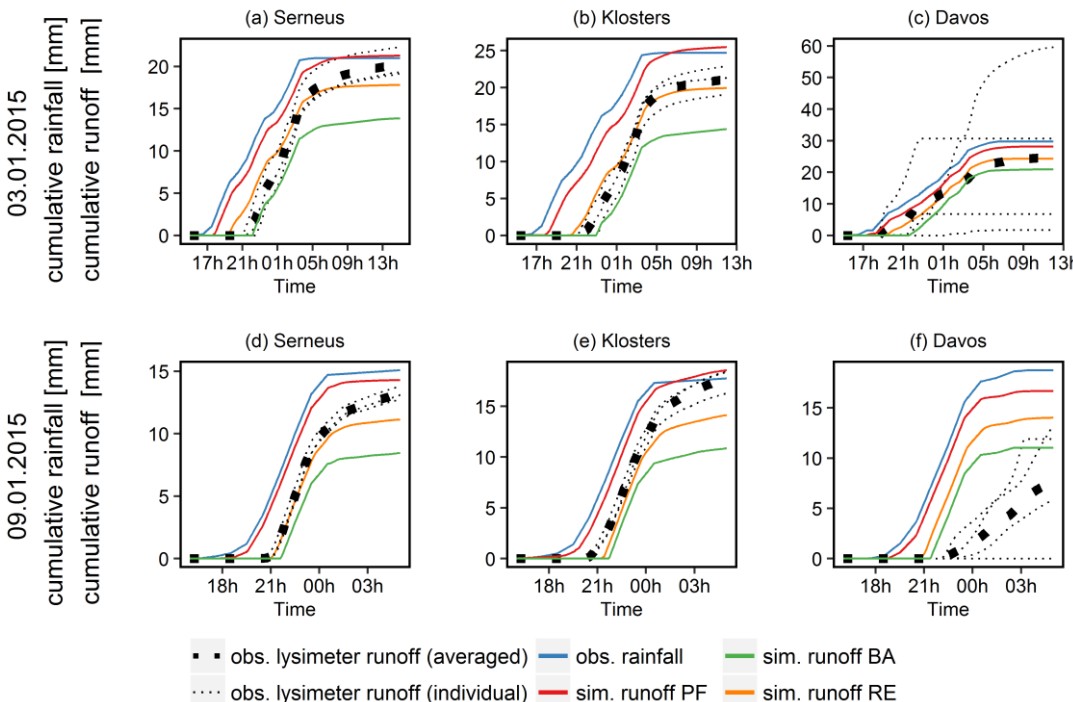

**Figure 6: Natural ROS events at 3<sup>rd</sup> and 9<sup>th</sup> of January 2015 in (a) Serneus, (b) Klosters and (c) Davos**

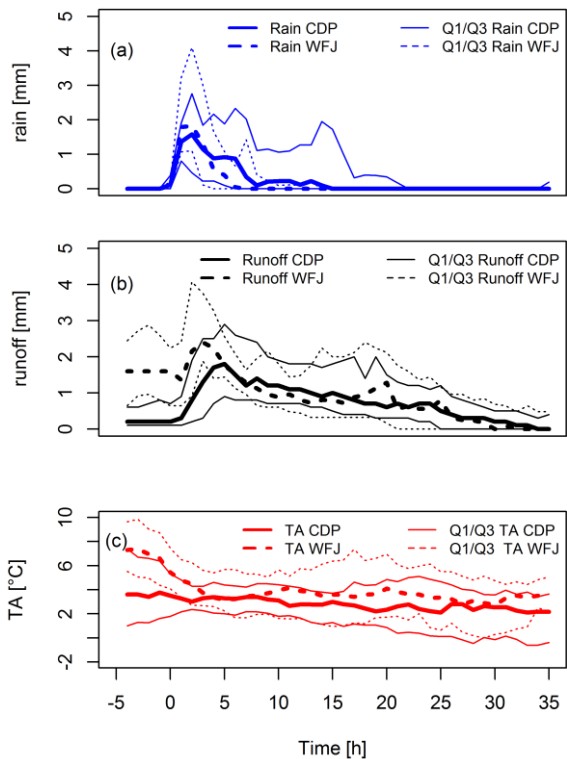

**Figure 7: (a) Temporal course of median rain (a), measured snowpack runoff (b) and air temperature (c) for WFJ (dotted) and CDP (solid) aggregated over all 40 and 61 events respectively. The thinner lines represent the lower and upper quartiles, respectively. The displayed period is extended by 5 hours prior to event beginning according to the event definition (0 h).**

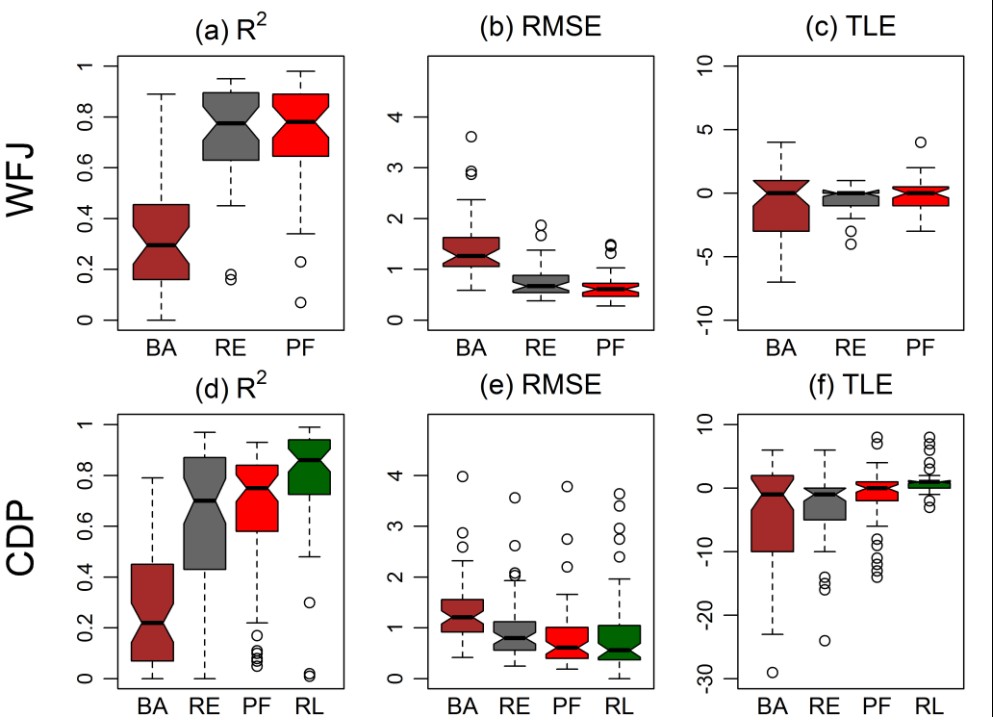

**Figure 8: RMSE, R$^2$ and TLE for simulations of 61 ROS events at the CDP site and of 40 ROS events at the WFJ site for all models (BA, RE, PF) and the reference lysimeter (RL) available only for CDP.**

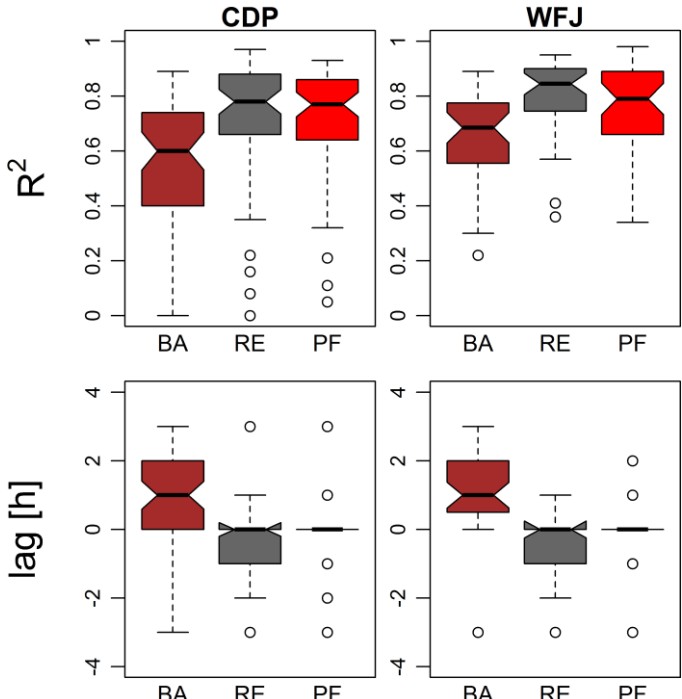

**Figure 9: Best R<sup>2</sup> values and corresponding lags using a cross-correlation function allowing a time shift (lag) of max -/+ 3 hours.**

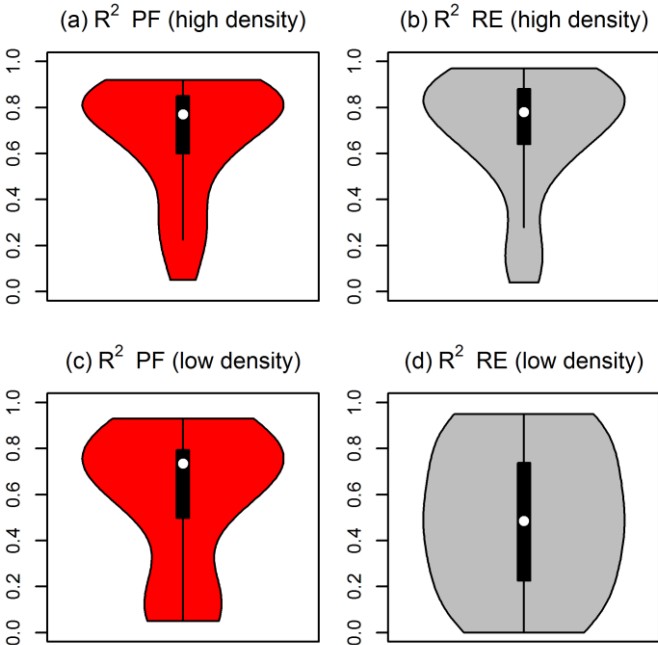

**Figure 10: Distribution of event-$R^2$ for CDP events for the PF (a,c) and RE (b,d) model. The sample is split into initial bulk snow densities above 350 kg m$^{-3}$ (a,b) and below 350 kg m$^{-3}$ (c,d).**