# Peer review of "Modeling liquid water transport in snow under rain-on-snow conditions – considering preferential flow"

_Hydrology and Earth System Sciences, 2016_

## Referee Comment (RC1) · Anonymous Referee #1 · 31 Aug 2016

The paper presents a new water transport scheme for the 1-D multi-layer physics-based SNOWPACK model that accounts for preferential flow effects. The model bases on a dual-domain approach and solves Richards equation for matrix and preferential flow. The area of fingers is explicitly parametrized using results from previously available laboratory experiments. Exchange of water between the matrix and the preferential domains is ruled either by water entry pressure head or by water saturation. The approach is evaluated using an extensive dataset of rain-on-snow events (ROS) from two different locations within European Alps and some field experiments. The proposed scheme demonstrates an improved performance at the scale of single ROS events and at the scale of a snow season.

[Figure]

Including preferential flow in snow models represents an important goal for snow hydrology. This is because it can provide an efficient routing of liquid water through snow and can generate snowmelt runoff earlier than expected. A frequent limitation for snow modelers is that the process understanding is still limited. In this regard, the paper proposes a parsimonious approach that parametrizes the portion of area occupied by fingers and thus takes this process into account without using a full 3-D geometry. The evaluation strategy is extensive and thorough and the paper is generally well written. I have some suggestions for authors that may be included with little effort. I can therefore suggest publication of the paper pending some (minor) revision.

My main suggestion regards Section 3 (Results) and 4 (Discussion). While I generally found both sprinkling experiments and the focus on long-term datasets well motivated and discussed, I am unsure that the two natural ROS events will provide a specific insight into this evaluation. Discussing some "real-world" applications is clearly important, but authors already do that using around 100 ROS events from Davos and Col de Porte. Moreover, results are "partly contradictory" when compared to artificial ROS simulations and this may be understandable as the physics is complex and data may be noisy. This is why focusing on a larger number of events (Section 3.3) is clearly more meaningful. So I suggest that either authors elaborate on the implications of these two specific events, or they remove Section 3.2, move this focus in the Discussion and use it as a starting point for discussing future research.

In the Discussion, I would also try to comment a bit more extensively on the dual-domain approach. For example, Eq. 1 relates the area of preferential flow to grain radius, which is for sure the most important variable ruling heterogeneity of water in snow. Because experimental observations of this process are still limited, may you suggest some directions for future research in order to improve this parametrization? May it also depend on supply rate or other conditions of the snow? More importantly, the model includes a parameter that needs to be calibrated. While calibration is helpful to compensate for a lack of physical understanding (and this is definitely the case with

preferential flow), it may be interesting for other users to know how did you choose the value of this parameter, or which would be the best calibration protocol for it. This is especially important where lysimeter data are not available. Which is the sensitivity of your results on the value of this parameter?

Specific comments: - Abstract: I found lines 11 – 17 a bit wordy. Could you try to summarize this? Furthermore, I would also specify the meaning of "balanced" (line 24) as it may be unclear for diagonal readers who are not going to screen the entire text;

- Line 29 page 2: may "capillary gradients" work better than "capillarity" alone?

- Line 17 page 3: remains -> remain?;

- Line 20 page 4: may authors include a brief comment about the reason why snow depth is constrained to observed values in a hydrologic application?

- Section 2.2: I would probably be more explicit about the simulated effect of preferential flow on water velocity. In my understanding, the model accelerates liquid water flow in snow because it concentrates water mass in small fingers where unsaturated conductivity is larger than in the matrix domain (and where refreezing is not allowed). Is this a correct interpretation? If yes, I would write something similar in the text in order to clarify this point.

- Eq. 1: should the exponent be negative as in Wever et al. (2016) on TCD?

- Line 19 – 20 page 6: may authors clarify which features of the sprinkler make it "especially developed for sprinkling on snowpack"?

- Line 18 and Table 1: did you choose different portions of snowpack for your experiments at the same sites?

- Line 27 page 7: I think including cumulative plots in Fig. 2 may definitely help to understand this methodology;

- Line 28 page 8: is this Table 1 instead of 2?

- Section 3.3.2: may you include some additional information about the observed variance of these plots? This may be helpful to put these lines in context;

- Lines 4 – 22 page 11: I found this paragraph a bit difficult to read. Could you please try to rephrase it and try to reorganize the information around the most important findings? This is a key step in the paper and therefore I think it should be very clear.

- Lines 24 – 28 page 11: which is the temporal resolution of lysimeter data? May this temporal resolution play a role for this analysis?

- Line 28 page 12: may refreezing be another important process here? This may be also important at lines 6 – 15 page 13.

- Lines 1 – 6 page 14: Katsushima et al. 2013 used a limited range of snow density in their experiments, and this range mostly includes densities greater than 380 kg/m3. May this help to explain this correlation?

- Figure 2: may the bar be above the x-axis?

- Figures 3, 4, etc.: could authors use different colors for the PF or BA approaches? In these figures, they are very similar and this is not very clear;

- Figure 6: is measured runoff black instead of red?

---

## Referee Comment (RC2) · Anonymous Referee #2 · 12 Sep 2016

Implementation of preferential flow process into one-dimensional model is challenging and important research. Accuracy of hydrological process will be enhanced by this improvement. Concepts and mechanisms of dual domain approach are described in more detail in the companion paper, Wever et al. (2016). Therefore, the main focus of this paper is the validation of the preferential flow model in terms of accuracy of runoff simulation. In this paper, authors performed the comparison with field data and showed the enhancement of accuracy in runoff by implementation of dual domain approach. The product of this study is appropriate to publish for HESS. On the other hand, although many contents of this paper described the success of this improvement, detailed analysis of the improved results are not sufficient. For example, information

of snowpack was not shown and not considered in the discussion despite it affects significantly of the relationship between water supply and runoff. Information of snow stratigraphy helps to understand when and why PF model obtained better accuracy than RE model. In this study, authors used the SNOWPACK model. Therefore, it is not difficult to add the snowpack information. If there are observed data by snow pit observation, showing observed data is desirable. As well as showing snow stratigraphy, quantitative estimation of snowmelt amount is also necessary to discuss runoff as a response of the liquid water input. It can also be estimated from the output of the SNOWPACK model.

Minor comments

P5 L18: This sentence describes Equation (1) is determined by Katsushima et al. (2013) and field observation data. Can you add the data used here (field observation data) in this paper? If it is already shown in previous paper, it should be referenced.

Fig 1 Please indicate the position of the sections of Fig. 1 (b) and (c) in the Fig. 1 (a)

P8 L16 Snowmelt amount should be considered in the analysis. It can be estimated from output data of the SNOWPACK.

Fig 3-5 Information of snow stratigraphy had better be added in these figures because it affects the relationship between input water supply and runoff. Although snow depth, averaged snow temperature and water content are shown in Table 2, it is not sufficient because water infiltration process is affected by more complicated snow conditions such as existence of ice layer, grain size contrast and ratio of wet snow to dry snow.

p9 L10-12 I guess that the reason of grater variability of snowpack runoff in highest located site is the existence of lateral flow due to ice layer or capillary barrier. In snowpack observation, are ice layer or capillary barrier existed?

P10 L20 This sentence indicated that snowmelt affected runoff significantly. Therefore, snowmelt amount should be estimated. Analysis considering snowmelt amount will

make better discussion.

P11 L4 RMSE(d,e) -> RMSE (b,e)

P11 L10 R2 value in WFJ was improved by PF model more significant than that in CDP. This result implies preferential flow predominated more in WFJ. Does the ratio of dry snow in WFJ was larger than that in CDP?

P12 In the discussion section, success of PF model was discussed mainly. However, discussions about shortcomings of the model and suggestion of required improvement are also important for future research.

P14L1 This sentence indicated that snow densities were spread from below 200kg m-3 up to 500 kg m-3 in CDP. But the accuracy of hydrological parameters (e.g. suction and hydraulic conductivity) of low-density snow in numerical snowpack model are insufficient because measurement of them is difficult. They are estimated using equations formulated based on the measured results using high-density snow. For this reason, inadequate accuracy is anticipated when low-density snow comprises a portion of snowpack. Can you add the discussion about the accuracy of runoff simulation in the case of snowpack with low-density snow? It may provide the information whether hydrological parameters of low-density snow should be measured in some way or is not important for runoff estimation.

P14 L30 Do you have any suggestion to improve the model? The companion paper, Wever et al., suggested some ideas to enhance the accuracy of ice layer formation. Suggestion to enhance the accuracy of runoff is welcome in this paper. Discussions considering snow stratigraphy help to provide idea for further improvement.

---

## Author Response (AR1)

Dear Prof. Eng. Carlo De Michele,

We would like to thank you and both reviewers for their helpful comments which certainly helped to improve this study. Please find our discussion of the revisions we have made in response to the comments from the two reviewers on the following pages. The track-changes version of the manuscript with all changes marked in red is found at the end of this reply letter. Please note that the line numbers given in the specific replies refer to the original manuscript, whereas the references to changes made in the manuscript refer to the revised manuscript below.

During the revisions we decided to alter the representation of the soil column for WFJ as we discovered instances of ponding on the top soil layer which would constrain snowpack runoff. To allow better comparison with snow lysimeter data we now ensure that infiltration into the soil column is never blocked. Such instances of ponding were only observed for WFJ and with the RE model, consequently only performance metrics related to this model / site changed noteworthy (improved). This along with additional analysis as suggested by reviewer 2 provided better and more consistent insights into the performance of the different models as they relate to snow properties and its stratigraphy.

We are grateful for the interesting reviews and comments that in our opinion enabled to improve the manuscript considerably.

Thank you and with best regards,

Sebastian Würzer (on behalf of the authors)

**Reply to general comments of Reviewer 1:**

The paper presents a new water transport scheme for the 1-D multi-layer physics based SNOWPACK model that accounts for preferential flow effects. The model bases on a dual-domain approach and solves Richards equation for matrix and preferential flow. The area of fingers is explicitly parametrized using results from previously available laboratory experiments. Exchange of water between the matrix and the preferential domains is ruled either by water entry pressure head or by water saturation. The approach is evaluated using an extensive dataset of rain-on-snow events (ROS) from two different locations within European Alps and some field experiments. The proposed scheme demonstrates an improved performance at the scale of single ROS events and at the scale of a snow season.

Including preferential flow in snow models represents an important goal for snow hydrology. This is because it can provide an efficient routing of liquid water through snow and can generate snowmelt runoff earlier than expected. A frequent limitation for snow modelers is that the process understanding is still limited. In this regard, the paper proposes a parsimonious approach that parametrizes the portion of area occupied by fingers and thus takes this process into account without using a full 3-D geometry. The evaluation strategy is extensive and thorough and the paper is generally well written. I have some suggestions for authors that may be included with little effort. I can therefore suggest publication of the paper pending some (minor) revision.

My main suggestion regards Section 3 (Results) and 4 (Discussion). While I generally found both sprinkling experiments and the focus on long-term datasets well motivated and discussed, I am unsure that the two natural ROS events will provide a specific insight into this evaluation. Discussing some "real-world" applications is clearly important, but authors already do that using around 100 ROS events from Davos and Col de Porte. Moreover, results are "partly contradictory" when compared to artificial ROS simulations and this may be understandable as the physics is complex and data may be noisy. This is why focusing on a larger number of events (Section 3.3) is clearly more meaningful. So I suggest that either authors elaborate on the implications of these two specific events, or they remove Section 3.2, move this focus in the Discussion and use it as a starting point for discussing future research.

In the Discussion, I would also try to comment a bit more extensively on the dual domain approach. For example, Eq. 1 relates the area of preferential flow to grain radius, which is for sure the most important variable ruling heterogeneity of water in snow. Because experimental observations of this process are still limited, may you suggest some directions for future research in order to improve this parametrization? May it also depend on supply rate or other conditions of the snow? More importantly, the model includes a parameter that needs to be calibrated. While calibration is helpful to compensate for a lack of physical understanding (and this is definitely the case with preferential flow), it may be interesting for other users to know how did you choose the value of this parameter, or which would be the best calibration protocol for it. This is especially important where lysimeter data are not available. Which is the sensitivity of your results on the value of this parameter?

We thank the reviewer for his constructive comments and ideas to improve the manuscript. Below, we give our response to the issues raised by the reviewer.

We agree that one could question the benefit of including the two natural events into this evaluation (Section 3.2). However, only for these events we have a multi-lysimeter setup, which raises the awareness that the observed processes can show considerably spatial heterogeneity as e.g. documented in (Figure 6). Further, these events show some limitations of the model. In the original version of the manuscript this point may not have been stated clearly enough and we will consider using these events for opening the discussions section as a starting point for discussing limitations of the preferential model and further steps needed in improving the model, as recommended by the reviewer. A more detailed discussion about further research needs and limitations of the model was also requested by Reviewer 2.

**Changes: Because of the reasons stated above, we decided to keep the natural events in place. However, we extended the discussion about the value of the multi-lysimeter approach and the heterogeneity of runoff observations in section 4.**

As stated by the reviewer, the area of preferential flow (Eq. 1) is likely to also depend on the water supply rate. Data using sandy soils from Glass et al. (1989), shown in DiCarlo (2013), suggests that with increasing system influx rates (100cm/min), the finger width of preferential flow is increasing, whereas it stays small for lower fluxes (20cm/min). However we are not aware of any experiments that have determined the area of preferential flow in snow for influx rates that are typical for natural ROS events. This might lead to false assumptions concerning the area involved in preferential flow and number of fingers, which is in turn important for the refreeze process. Even though we have used the lowest influx rates from Katsushima et al. (2013), these might still lead to bias concerning the preferential flow area. More experimental data under conditions with lower influx rates would be desirable. This is described in Wever et al. (2016b) in detail and we will refer to this study more explicitly. Another source of inaccuracy for deriving the preferential flow area are the samples densities (all above 380 kg/m$^3$) used by Katsushima et al. (2013).

**Changes: We expanded the discussion about the role of system influx rates in section 4. See page 16 lines 3ff. in the revised manuscript.**

The calibrated parameters are related to a physical process (threshold for saturation and number of flowpaths). Ideally, they should not have to be calibrated for every model application but rather determined from laboratory experiments. We agree that the calibration of the parameters is an important part of the model, and therefore will refer to the study of Wever et al. (2016b) more explicitly, where this topic is discussed in greater detail. Here we would like to focus on the discussion about the role of rain-on-snow for preferential flow.

**Changes: For the calibration of parameters, we refer to Wever et al. (2016b). However, we decided to use a threshold in saturation for preferential flow (return flow condition) of 0.06, which has been determined to reproduce runoff best in the sensitivity study presented in Wever et al. (2016b), which is now described in section 2.2 on page 6 lines 8ff.**

**Reply to specific comments of Reviewer 1:**

Abstract: I found lines 11 – 17 a bit wordy. Could you try to summarize this? Furthermore, I would also specify the meaning of "balanced" (line 24) as it may be unclear for diagonal readers who are not going to screen the entire text.

Reply: We will rewrite this in the revised manuscript. The term „balanced" means that for the extensive dataset of WFJ and CDP, the interquartile range is smaller with the PF model and time lag errors are smaller.

**Changes: We revised the abstract in the revised manuscript, but mainly we decided to be more specific about the results and therefore replace the term "balanced".**

Line 29 page 2: may "capillary gradients" work better than "capillarity" alone?

Reply: Colbeck (1972) used the term capillarity. We amend the manuscript, changing „capillarity" to „capillary forces". We think that in this context having a general expression suits best.

**Changes: „capillarity" was replaced by „capillary forces" in the revised manuscript.**

Line 17 page 3: remains -> remain?;

Reply: It will be changed in the updated manuscript, thanks!

**Changes: "remains" was replaced by "remain" in the revised manuscript.**

Line 20 page 4: may authors include a brief comment about the reason why snow depth is constrained to observed values in a hydrologic application?

Reply: It is true that for Weissfluhjoch, a dataset with undercatch-corrected precipitation data is available. Nonetheless, because the timing of snowpack runoff is essentially dependent on the snow height, we wanted to exclude this potential source of error to achieve the best comparability between the 3 water transport models. Because we focus on the event-scale we constrained the simulations to the observed snow height, such that we have an accurate snow depth at the onset of the events.

**Changes: In the revised manuscript, we now justify the approach of constrained snow heights in section 2.1, page 4, lines 22ff.**

Section 2.2: I would probably be more explicit about the simulated effect of preferential flow on water velocity. In my understanding, the model accelerates liquid water flow in snow because it concentrates water mass in small fingers where unsaturated conductivity is larger than in the matrix domain (and where refreezing is not allowed). Is this a correct interpretation? If yes, I would write something similar in the text in order to clarify this point.

Reply: The model indeed accelerates liquid water flow in snow because it concentrates water mass in a smaller area where the saturation is hence higher and unsaturated conductivity is larger. This happens faster in the preferential domain, representing only a fraction of the snow cover. Additionally, refreezing is not taking place in the preferential domain in the current approach. We will add a clearer description in the updated manuscript.

**Changes: We added a short summary about the principle of the model as suggested by the reviewer in section 2.2, page 6, lines 14ff.**

Eq. 1: should the exponent be negative as in Wever et al. (2016) on TCD?

Reply: The exponent should be identical to the one presented in Wever et al. (2016), and this appears to be the case. If your pdf viewer shows different values it might be a technical error of the pdf document.

**Changes: No changes.**

Line 19 – 20 page 6: may authors clarify which features of the sprinkler make it "especially developed for sprinkling on snowpack"?

Reply: The sprinkled area and sprinkling intensity depend on the water pressure at the nozzle and the distance of the nozzle to the snow surface. The sprinkling device was calibrated using different pressures at the nozzle and distances to the surface to achieve preferably low intensities within naturally occurring range and at the same time a uniform distribution of sprinkling intensity over the lysimeter area. The device was developed to easily be able to adapt the sprinkling height to the height of the snow cover, so that the distance stays in the calibrated area. It is also lightweight to be able to move, set up the device and conduct the experiments within one day. This is crucial for being able to conduct the sprinkling experiments, but might be of smaller relevance for this study. We therefore decided to delete this sentence, as the device is already described in Juras et al. (2013) and add another reference (Juras et al., 2016).

**Changes: We changed the sentence according to our reply.**

Line 18 and Table 1: did you choose different portions of snowpack for your experiments at the same sites?

Reply: The multi-lysimeter setup (3-4 at each site) allowed us to use every lysimeter just once. Because they were installed before the first snowfall, the snowpack on the lysimeters was

undisturbed. In one case (Klosters, 8-Apr-2015) we used the same lysimeter twice, because the lysimeter became free of seasonal snow cover and the experiment was conducted on fresh snow which fell the day before.

**Changes: We revised the corresponding part in section 2.4 on page 7, lines 3ff. to clarify the use of lysimeters in the experimental procedure.**

Line 27 page 7: I think including cumulative plots in Fig. 2 may definitely help to understand this methodology;

Reply: We will replace Fig. 2 in the original manuscript by Fig. 1 in this response.

**Changes: The figure was extended by a cumulative plot in the revised manuscript.**

Line 28 page 8: is this Table 1 instead of 2?

Reply: Yes, sorry for causing confusion. This will be changed in the updated version of the manuscript. Thanks!

**Changes: The numeration has been changed in the revised manuscript**

Section 3.3.2: may you include some additional information about the observed variance of these plots? This may be helpful to put these lines in context;

Reply: See Fig. 2 in this reply letter. The original Figure will be replaced by a similar Figure.

**Changes: The figure was extended by the 1$^{st}$ and 3$^{rd}$ quartile of the corresponding values to provide some information about the observed variance.**

Lines 4 – 22 page 11: I found this paragraph a bit difficult to read. Could you please try to rephrase it and try to reorganize the information around the most important findings? This is a key step in the paper and therefore I think it should be very clear.

Reply: We will adapt the manuscript in this part. Thanks for the advice.

**Changes: We revised the paragraph and we think that the results are now described in a more understandable way.**

Lines 24 – 28 page 11: which is the temporal resolution of lysimeter data? May this temporal resolution play a role for this analysis?

Reply: The temporal resolution of the lysimeter data used for the extensive dataset in this study is 1 hour. The temporal resolution may clearly play a role for this analysis. Wever et al. (2014) have already shown that for comparing the BA and RE model, improvements by RE are found particularly for subdaily timescales, but are less important for daily sums. Especially $R^2$ values strongly depend on the correct representation of increasing or decreasing runoff at the given time step. In the revised manuscript, we will discuss the effect of the temporal resolution of the lysimeter by analysing the 30 minute lysimeter data from WFJ.

**Changes: In section 2.5, page 8, lines 4ff., we added information about resolution of lysimeter data. To assess the influence of the temporal resolution of the runoff data, we additionally compared simulations and observations at 30 minute time steps for WFJ, where meteorological and lysimeter data were available at higher temporal resolution. With similar values, this analysis was consistent with the results shown for the 1 hour resolution and we decided not to show the data.**

Line 28 page 12: may refreezing be another important process here? This may be also important at lines 6 – 15 page 13.

Reply: Indeed, refreezing should be discussed here. First, neglecting refreeze in the PF model leads to earlier runoff for the cold snow covers. However, the cold content should be consumed by the end of the event and therefore refreeze should not be accountable for the difference in total event runoff between the RE and PF model. This difference might be attributable to differences in water held in the capillaries. We still think that the main processes of overestimating total event runoff for the RE and PF model are underestimation of water held in the capillaries and high lateral flow, observed during the experiment for cold initial conditions. The effect of high lateral flow is also confirmed by SWE measurements before and after the experiments, which show little changes, being within normal spatial variability and measurement errors. Lateral flow likely led to an effective loss of sprinkling water per surface area of the lysimeter, which of course cannot be reproduced by the models. The short time lag for the 1st natural ROS event at Davos, even having the coldest snowpack, is contributing to the assumption that refreeze is limited in preferential flow paths. We will add this to the discussion. Thanks for the advice!

**Changes: We added a comment on the role of refreeze in section 4 on page 14 lines 9ff.**

Lines 1 – 6 page 14: Katsushima et al. 2013 used a limited range of snow density in their experiments, and this range mostly includes densities greater than 380 kg/m3. May this help to explain this correlation?

Reply: Indeed, this is a likely explanation for this correlation. We will add this to the discussion, as already stated in the reply to the major comments. This might also explain the bad representation of runoff for the natural events where densities ranged from 180-220 kg m$^{-3}$ on Jan 3rd and 250-310 kg m$^{-3}$ on Jan 9th. For the sprinkling events, densities were around 220-270 kg m$^{-3}$ for the winter experiments and 300-400 kg m$^{-3}$ for the spring experiments. Also here we see an improvement in runoff representation with density. See also Fig. 3 in this reply letter.

**Changes: We extended the discussion in section 4 on page 15 lines 18ff. on the role of the experimental data on determining the parameters used in the respective models and their potential implications on the respective model performances. Please consider that the Fig. 3 in this reply shows data from the original manuscript, whereas Fig. 10 in the updated manuscript contains data from new simulations.**

Figure 2: may the bar be above the x-axis?

Reply: Thanks! The caption will be changed accordingly.

**Changes: We changed the caption of the figure accordingly.**

Figures 3, 4, etc.: could authors use different colors for the PF or BA approaches? In these figures, they are very similar and this is not very clear;

Reply: The colours will be changed in all Figures with that problem.

**Changes: The colours have been changed in the corresponding figures of the revised manuscript.**

Figure 6: is measured runoff black instead of red?

Reply: Thanks! The caption will be changed accordingly.

**Changes: We changed the caption of the figure accordingly.**

**Figures:**

[Figure]

**Figure 1: (a) Example of a ROS event occurring at WFJ. The entire extent of the x-axis refers to the evaluation period; the bar above the x-axis refers to the event length. (b) Cumulative version of the plot.**

[Figure]

**Figure 2: Course of median rain (a), measured snowpack runoff (b) and air temperature (c) for WFJ (dotted) and CDP (solid) for all 40 and 61 events respectively. The thinner lines represent the lower and upper quartiles, respectively. The displayed period is extended by 5 hours prior to event beginning according to the event definition (0 h).**

[Figure]

**Figure 3:** $R^2$ values for CDP events for the PF (a,c) and RE (b,d) model. The sample is split for bulk densities above 350 kg m$^{-3}$ (a,b) and below 350 kg m$^{-3}$ (c,d).

**Reply to general comments of Reviewer 2:**

Implementation of preferential flow process into one-dimensional model is challenging and important research. Accuracy of hydrological process will be enhanced by this improvement. Concepts and mechanisms of dual domain approach are described in more detail in the companion paper, Wever et al. (2016). Therefore, the main focus of this paper is the validation of the preferential flow model in terms of accuracy of runoff simulation. In this paper, authors performed the comparison with field data and showed the enhancement of accuracy in runoff by implementation of dual domain approach. The product of this study is appropriate to publish for HESS. On the other hand, although many contents of this paper described the success of this improvement, detailed analysis of the improved results are not sufficient. For example, information of snowpack was not shown and not considered in the discussion despite it affects significantly of the relationship between water supply and runoff. Information of snow stratigraphy helps to understand when and why PF model obtained better accuracy than RE model. In this study, authors used the SNOWPACK model. Therefore, it is not difficult to add the snowpack information. If there are observed data by snow pit observation, showing observed data is desirable. As well as showing snow stratigraphy, quantitative estimation of snowmelt amount is also necessary to discuss runoff as a response of the liquid water input. It can also be estimated from the output of the SNOWPACK model.

> We thank the Reviewer 2 for his constructive comments and ideas to improve the manuscript. Below, we give our response to the issues raised by the reviewer.

> We agree that the snow stratigraphy is a very important factor influencing liquid water transport in the snowpack. It certainly can help understanding when and why the PF model obtained better accuracy than RE model. Eiriksson et al. (2013) for example stated the importance of ice layers and stratigraphic boundaries for runoff formation at the slope scale. Considering findings of Wever et al. (2016), we think that even though considering preferential flow enabled to simulate ice layers, the probability of detection is still not sufficient to analyse the effect of ice layers on runoff generation. So far, we focused on bulk snow cover properties derived by the simulations and their effect on the model performance, as shown in the discussion section in the original manuscript for snow height and bulk density. Additionally, Wever et al. (2016) show the good representation of density observations of RE and PF model at WFJ. We therefore plan to investigate the existence of marked grain size and density changes, representing possible capillary barriers, on the performance of the different models. Also for the experiments and natural events, snow stratigraphy information will be discussed, as far as available.

**Changes: A paragraph about stratigraphic boundaries representing e.g. possible capillary barriers was added to the discussion on page 15 lines 18ff.**
* * *
> It is true that snowmelt is an important source of liquid water input for runoff generation. The total influx rate, consisting of both, rainfall and snowmelt can, besides snowpack stratigraphy, be an additional source of variability in the performance of the preferential model, since the area involved in preferential flow has been shown to be dependent on water influx rates for laboratory experiments in soil physics (DiCarlo, 2013). We will consider this in the updated manuscript.

**Changes: The snowmelt amounts were very similar for all models for the extensive dataset and therefore cannot explain the different performance of the respective models. However, a paragraph about the influence of liquid water input rates on the model performance for the extensive dataset was added to the discussion on page 16 lines 14ff. The snowmelt amounts for the sprinkling experiments were added to the manuscript on page 9, lines 2ff and 6. For the Davos field site natural events, the differences between cumulative snowpack runoff of the respective models are mainly due to the different amounts of liquid water stored in the snowpack, which are considerably lower in case of the PF model.**

**Reply to specific comments of Reviewer 2:**

P5 L18: This sentence describes Equation (1) is determined by Katsushima et al. (2013) and field observation data. Can you add the data used here (field observation data) in this paper? If it is already shown in previous paper, it should be referenced.

> Reply: The data used from Katsushima et al. (2013) is presented in a graph in Wever et al. (2016b). The field observation data was actually not used for the fit in the end and we will delete that part of this sentence.

**Changes: The sentence has been adapted as stated in the reply.**

Fig 1 Please indicate the position of the sections of Fig. 1 (b) and (c) in the Fig. 1 (a)

> Reply: Apparently we caused confusion here, as all the figures show different experiments and don't relate to each other. We will revise the caption. (b) relates to Serneus 26.02.15 and (c) relates to the experiment in Klosters on 26.03.15.

**Changes: The caption for the figure was changed in the revised manuscript.**

P8 L16 Snowmelt amount should be considered in the analysis. It can be estimated from output data of the SNOWPACK.

> Reply: In general, liquid water input and therefore snowmelt represents a substantial amount of liquid water input and is a key factor influencing the water transport. We will estimate the snowmelt from the simulation output. As stated in our reply to the major comments, we also plan to extend the discussion in this sense.

**Changes: The snowmelt amounts derived by SNOWPACK simulations have been added to the results of the sprinkling experiments on page 9 line 6.**

Fig 3-5 Information of snow stratigraphy had better be added in these figures because it affects the relationship between input water supply and runoff. Although snow depth, averaged snow temperature and water content are shown in Table 2, it is not sufficient because water infiltration process is affected by more complicated snow conditions such as existence of ice layer, grain size contrast and ratio of wet snow to dry snow.

> Reply: As stated in the reply to major comments, we can extent the analysis by considering snowpack stratigraphy. In Fig. 1 of this reply letter, we provide the temperature and LWC profiles including identified crusts and ice layers for the sprinkling experiments. Unfortunately, no detailed information about grain size and shape is available for the snow profiles.

**Changes: A Figure showing observed profiles of liquid water content, snow temperature and ice layers and crusts for the sprinkling experiments was added to the manuscript (Figure 3).**

p9 L10-12 I guess that the reason of greater variability of snowpack runoff in highest located site is the existence of lateral flow due to ice layer or capillary barrier. In snowpack observation, are ice layer or capillary barrier existed?

> Reply: Unfortunately, no detailed information about snow microstructure stratigraphy is available for the natural events. However, ice layers and the position of layer transitions were qualitatively assessed before the onset of rainfall. For the Event of Jan 03, no distinct ice layers were observed. The snow cover built up within the previous week with mostly very cold temperatures and just a light melt refreeze crust at the top. For the Jan 09 event, the previous event lead to distinct ice layers. But apparently, these differences in the snowpack layering for both events are not expressed by different behaviour of the lysimeter. They are in both cases very heterogeneous in runoff.

**Changes: We added a comment about observed ice layers to the results and also discuss the spatial heterogeneity of measures runoff in the discussion section 4 on page 14 lines 21ff.**

P10 L20 This sentence indicated that snowmelt affected runoff significantly. Therefore, snowmelt amount should be estimated. Analysis considering snowmelt amount will make better discussion.

> Reply: As stated in our reply to the major comments, we plan to extend the discussion in this sense and provide information about snowmelt. We plan on analysing snowmelt as an important part of system input rates and analysing their implication on the performance of the water transport models.

**Changes: We analysed the snowmelt contribution during the events. The snowmelt amounts of the respective simulations are very similar and therefore cannot explain the different performance of the models. However, we added a discussion about the effect of water input rates (sum of rain rates and snowmelt rates) on the model performance on page 16 lines 3ff.**

P11 L4 RMSE(d,e) -> RMSE (b,e)

Reply: This will be changed in the updated manuscript. Thanks!

**Changes: This has been changed in the revised manuscript**
* * *
P11 L10 R2 value in WFJ was improved by PF model more significant than that in CDP. This result implies preferential flow predominated more in WFJ. Does the ratio of dry snow in WFJ was larger than that in CDP?

Reply: By applying a bulk threshold of 1 vol% LWC to separate between dry and wet initial snow conditions, 30% of events at CDP had a dry initial snow cover, whereas this was the case for just 1 event at WFJ. If looking at the ratio of wet layers to dry layers within the snowpack, this single event had 15% of the layers wet, whereas for all other events at least 99% of layers had a LWC of at least 1 vol%.

**Changes: The decision to alter the representation of the soil column for WFJ led to improved performance metrics related to the RE model at WFJ. In the new simulations we show that the difference in PF and RE model performance is rather little for WFJ and in fact bigger for CDP, where we also observe a bigger variability in snowpack properties. The discussion has been extended concerning the effect of snowpack stratigraphy measures having possible implications on the singe event-$R^2$ on page 15 lines 22ff.**
* * *
P12 In the discussion section, success of PF model was discussed mainly. However, discussions about shortcomings of the model and suggestion of required improvement are also important for future research.

Reply: Thanks for the advice. We will answer this with the last comment (P14, L30) in detail.

**Changes: The discussion on page 16 lines 14ff. has been extended concerning this point.**
* * *
P14L1 This sentence indicated that snow densities were spread from below 200kg m-3 up to 500 kg m-3 in CDP. But the accuracy of hydrological parameters (e.g. suction and hydraulic conductivity) of low-density snow in numerical snowpack model are insufficient because measurement of them is difficult. They are estimated using equations formulated based on the measured results using high-density snow. For this reason, inadequate accuracy is anticipated when low-density snow comprises a portion of snowpack. Can you add the discussion about the accuracy of runoff simulation in the case of snowpack with low-density snow? It may provide the information whether hydrological parameters of low-density snow should be measured in some way or is not important for runoff estimation.

Reply: Thanks for this comment and advice. This topic was also raised by Reviewer 1. The laboratory experiments from Yamaguchi et al. (2012) and Katsushima et al. (2013) were conducted on snow with densities of 380 kg m$^{-3}$ and above (typically 400-600 kg m$^{-3}$), therefore much more in range of the densities at WFJ (around 450-500 kg m-3). This could explain the higher variance in $R^2$ values for runoff at CDP (including densities below 200 kg m-3).  We will

adapt the discussion accordingly. This suggests that the variable performance of RE and PF models at CDP may be associated with the existence of lower snowpack densities. Fig. 2 in this reply letter shows $R^2$ values for the CDP-events, split in two samples with initial densities below and above 350 kg m$^{-3}$, respectively. The PF model shows better performance for lower densities when compared to the RE model. In general, the low-density parameters need more experimental backing – as the reviewer suggested - and we will discuss this adequately.

**Changes: A discussion of the variable model performance depending on snow density is now added on page 15 lines 15ff. Please consider that the Fig. 2 in this reply shows data from the original manuscript, whereas Fig. 10 in the updated manuscript contains data from new simulations.**

P14 L30 Do you have any suggestion to improve the model? The companion paper, Wever et al., suggested some ideas to enhance the accuracy of ice layer formation. Suggestion to enhance the accuracy of runoff is welcome in this paper. Discussions considering snow stratigraphy help to provide idea for further improvement.

Reply: The suggestions provided in Wever et al. (2016) for improving the preferential flow model apply also for this study. This concerns especially the two parameters which have been calibrated: the threshold for saturation ($\Theta_{th}$) and the number of preferential flow paths for refreeze (N). Laboratory experiments or detailed simulations using multi-dimensional snowpack models might be able to determine the number and size of preferential flow paths for lower input intensities and snow densities. Additionally, we think that very limited data of high temporal and spatial resolution snowpack runoff measurements are available, limiting validation possibilities. CDP and WFJ provide long-term measurements on an adequate temporal resolution; however, this data gives little information about spatial variability of snowpack runoff. Large area multi-compartment lysimeter setups might help improving estimating size, amount and spatial heterogeneity of flow fingers. The lysimeters should be at least 10 m$^2$ in total area to minimize the effect of preferential flow for the total lysimeter area (Kattelmann, 2000). Sprinkling experiments with low sprinkling intensities on such a device could fill a knowledge gap about water transport in snow under conditions naturally occurring, but under controlled conditions. Also the role of ice layers for vertical water transport is not yet resolved. During the sprinkling experiments in this study, the dry and cold snowpack showed highest lateral flows, but also clearly dominating vertical preferential flow. Our experiments for a ripe snowpack did not show pronounced lateral flow but distinct matrix flow, while Eiriksson et al. (2013) observed lateral flow in saturated layers during wet conditions on a slope. The implications of preferential flow or distinct lateral flow on ice layers or structural transitions can also be observed on the catchment scale. Rössler et al. (2014) had to adjust parameters, leading to very fast overland flow to be able to model the hydrological implications of a major ROS event in October 2011 in Switzerland. It is quite likely that this parameter had to be set that high to compensate for neglecting lateral flow within the snowpack or vertical preferential flow. To be able to better forecast such events, research has to be promoted from experimental lab and field studies over sophisticated multi-dimensional water transport models to simplified but operationally applicable 1D water transport models. We will amend the discussion in the manuscript regarding this point.

**Changes: The discussion on page 16 lines 14ff. has been extended concerning current limitations and suggestions how to possibly enhance the accuracy of runoff are now provided.**

**References:**

[revised manuscript text omitted]
 via RE to the PF model. For CDP, a distinct increase in R$^2$ values could be observed between BA and both RE and PF which showed a similar median R$^2$. The interquartile range of R$^2$ values was generally higher than for WFJ and increased from BA to RE, whereas it was decreasing for PF. Also the RMSEs significantly decreased with RE and PF, compared to the BA model. Similarly to WFJ, the median TLE for CDP was zero, except in the case of RE, where negative median TLE indicates that modelled snowpack runoff was on average a bit delayed compared to the observations. Nevertheless, the PF model showed the most consistent results, whereas the BA model showed the largest spread in TLE for individual events. The magnitude of TLE was generally higher for CDP than for WFJ and mostly negative, which means that the modelled snowpack runoff was delayed compared to lysimeter snowpack runoff. For BA and PF, TLE was more often positive (early modelled snowpack runoff), which led to a very good median for BA, but also a larger interquartile range. The PF model led to the same median as the RE model, but showed the smallest interquartile range. As reference 
[revised manuscript text omitted]